



# Highly-controlled, reproducible measurements of aerosol emissions from African biomass combustion

Sophie L. Haslett[1], J. Chris Thomas[2], William T. Morgan[1], Rory Hadden[2], Dantong Liu[1], James D. Allan[1,3], Paul I. Williams[1,3], Keïta Sekou[4], Cathy Liousse[5] and Hugh Coe[1].

[1] Centre for Atmospheric Science, University of Manchester, Manchester, M13 9PL, United Kingdom
[2] School of Engineering, University of Edinburgh, Edinburgh, EH9 3JL, United Kingdom
[3] National Centre for Atmospheric Science, University of Manchester, Manchester, M13 9PL, United Kingdom
[4] L'Université Félix Houphoët-Boigny, VPV34, Abidjan 01, Côte D'Ivoire
[5] Laboratoire d'Aérologie, Université Paul Sabatier Toulouse III, France

*Correspondence to*: Hugh Coe (hugh.coe@manchester.ac.uk)

**Abstract.** Particulate emissions from biomass burning can both alter the atmosphere's radiative balance and cause significant harm to human health. However, due to the large effect on emissions caused by even small alterations to the way in which a fuel burns, it is difficult to study particulate production of biomass combustion mechanistically and in a repeatable manner. In order to address this gap, in this study, small wood samples sourced from Côte

D'Ivoire in West Africa were burned in a highly-controlled laboratory environment. The shape and mass of samples, available airflow and surrounding thermal environment were carefully regulated. Organic aerosol and refractory black carbon emissions were measured in real time using an Aerosol Mass Spectrometer and a Single Particle Soot Photometer, respectively. This methodology produced remarkably repeatable results, allowing aerosol emissions to be mapped directly onto different phases of combustion. Emissions from pyrolysis were visible as a distinct phase

before flaming was established. After flaming combustion was initiated, a black-carbon-dominant flame was observed during which very little organic aerosol was produced, followed by a period that was dominated by organic-carbon-producing smouldering combustion, despite the presence of residual flaming. During pyrolysis and smouldering, the two phases producing organic aerosol, distinct mass spectral signatures that correspond to previously-reported variations in biofuel emissions measured in the atmosphere are found. Organic aerosol emission

factors averaged over an entire combustion event were found to be representative of the time spent in the pyrolysis and smouldering phases, rather than reflecting a coupling between emissions and the mass loss of the sample. Further exploration of aerosol yields from similarly carefully controlled fires and a careful comparison with data from macroscopic fires and real-world emissions will help to deliver greater constraints on variability of particulate emissions in atmospheric systems.

## 30  1 Introduction

Atmospheric aerosol particles emitted from biomass burning have a substantial influence on global climate, atmospheric chemistry and cloud processes, in addition to being detrimental to human health. Domestic fires used for cooking and heating, agricultural burning, forest wildfires and savannah fires all contribute to the atmospheric loading of biomass burning aerosol (BBA). Andreae & Merlet (2001) identify biomass burning to be the largest


source of primary fine carbonaceous particles in the atmosphere globally. Eighty per cent of global BBA is thought
to originate from the tropics (Hobbs et al., 1997) and a significant proportion of this is anthropogenic. However,
with the resurgence of biomass burning as a source of renewable energy in the developed world (Johansson et al.,
2004), issues surrounding BBA are of increasing concern worldwide.

Black carbon and organic carbon are both produced in high quantities by biomass burning, along with inorganic
species such as sulphates, nitrates and potassium. Black carbon absorbs incoming short-wave radiation, re-emitting
it at infrared frequencies that warm the surrounding air. In contrast, the majority of organic carbon and inorganic
species scatter incoming sunlight. Given the presence of both absorbing and scattering aerosol, it is clear that BBA
must be well characterised in order for its impact on climate to be fully understood. The ratio of organic matter to
black carbon is likely to have a large influence on the net effect of BBA in the atmosphere.

One of the greatest anthropological sources of black carbon is the residential burning of solid fuels, which
contributes 25% of emissions: more than both diesel engines and industrial coal burning (Bond et al., 2013). Given
this large contribution, it is important to be able to constrain the estimate of emissions from this source in order to
model black carbon in the Earth's atmosphere accurately. A considerable number of studies have been carried out in
order to establish the emission factors – the mass emitted per unit fuel burned – of various chemical species for
different fuels and circumstances. There has been a particular focus in the literature on emission factors from
cookstoves (e.g. MacCarty et al., 2008; Roden et al., 2009; Zhang et al., 2000) and wildfires (e.g. Christian et al.,
2007; Yokelson et al., 2008). Others, including Schneider et al. (2006), Weimer et al. (2008) and Elsasser et al.,
(2013) have investigated the chemical composition of BBA emissions in real time during biomass combustion in the
laboratory. Attempts to compile all available measurements of emission factors for a number of different species,
both gaseous and particulate, have been carried out by Andreae & Merlet (2001) and more recently by Akagi et al.
(2011).

However, due to the complex nature of biomass combustion, and high sensitivity to the combustion environment
(i.e. ventilation and heat transfer), it has proven difficult to establish repeatability in the results of laboratory-based
biomass burning experiments. Even small alterations in the burning environment can lead to huge variations in
particulate emission factors (Akagi et al., 2011) and as such, the majority of studies have found considerable
differences in the observed aerosol emission factors and efficiency of combustion between nominally identical
burns. Furthermore, the standard approach in many such studies has been to consider only average emission factors
from an entire combustion event, which can mask significant changes in emissions that take place throughout
combustion. Nonetheless, some recent studies have been carried out, which consider real-time particulate emissions
under more controlled conditions, including Alfarra et al. (2007) and Elsasser et al. (2013).

In ambient studies that measure the mass spectra of organic aerosols, techniques such as positive matrix factorisation
(PMF) are regularly used to identify and quantify distinct mass spectra with different properties (Zhang et al. 2011).
The multilinear engine (ME-2) uses pre-identified mass spectra to constrain the factors identified by PMF (Paatero,



1999; Canonaco et al., 2013). Biomass burning is known to produce one of the most difficult organic signatures to constrain using this method. This is due to its large variability. For example, Crippa et al. (2014) advise allowing for a variability of of $\pm$ 30-50% when attempting to constrain mass spectra from the biomass burning component of atmospheric aerosols in studies using the Aerosol Mass Spectrometer. In comparison, the recommended constraint for hydrocarbon-like OA is only $\pm$ 5-10%. An improved understanding of biomass burning on a mechanistic level has the potential to tighten these constraints.

In this study, an emphasis was placed on enhancing the repeatability of combustion events in order to study aerosol emissions on a mechanistic level, rather than attempting to simulate real-world combustion. In order to achieve this, wood samples were prepared to be as similar as possible during each test. The Fire Propagation Apparatus (ASTM, 2000), was used to control the combustion environment. Samples were heated by a series of infrared heaters and controlled air flow was delivered from below the sample.

Measurements of the combustion products were made in real time using an Aerosol Mass Spectrometer (AMS) (Aerodyne Research, Inc., Billerica, MA, USA) and a Single Particle Soot Photometer (SP2) (Droplet Measurement Technologies, Longmont, CO, USA) to measure organic aerosol (OA) and refractory black carbon (rBC) respectively. Emission factors were measured with a time resolution of 5 s, which allowed extremely fine detail in the dynamics of emissions to be distinguished.

Large differences were observed in the ratio between OA and rBC during different periods of each individual burning event. As such, three different phases of combustion were defined based on this ratio, each of which displayed a unique chemical signature.

## 2 Method

Combustion experiments were carried out in the Rushbrook Fire Laboratory, School of Engineering, University of
Edinburgh. The Fire Propagation Apparatus (FPA) allows determination and quantification of material flammability characteristics including time to ignition, gaseous emissions, burning rate and heat release rate. Small (approximately 100 mm) samples of fuel are heated and ignited under highly-controlled conditions (Brohez et al., 2006). A custom-made stainless steel basket was used to hold the samples and allowed air to flow them from beneath. A flow controller allowed control over the quantity of air reaching the sample from below; here, this was
held at either 50 or 200 lpm. The sample holder was placed on a mass balance. Four infrared heat lamps provided a spatially uniform heat flux across the surface of the samples . Heat fluxes of 30 kW m$^{-2}$ or 50 kW m$^{-2}$ were used. The sample was surrounded by a quartz tube to ensure that ambient laboratory air did not reach the fuel during combustion. A pilot flame was used to initiate flaming combustion, which had a negligible influence on CO and $CO_2$ emissions.




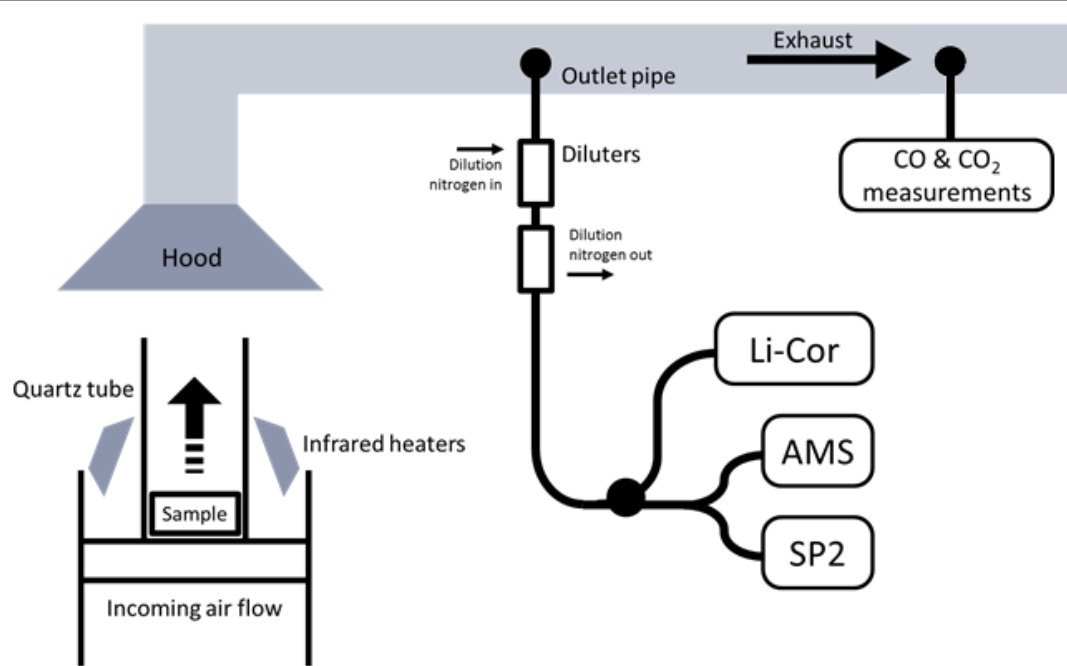

**Figure 1: The experimental set-up, showing the sample in the FPA and the location of the sample pipes in the exhaust.**

Exhaust gases from combustion were collected in a hood before entering into the exhaust duct, where turbulence

encouraged mixing. Air samples for aerosol measurement were extracted from the exhaust tube from a forward-

facing subsampling inlet. After passing through a filtering condensing system, concentration of oxygen, carbon

dioxide and carbon monoxide in the exhaust gas were measured using non-dispersive infra red techniques (Servopro

4200). A separate sample system allowed aerosol properties in the exhaust to be monitored in real time by the AMS

and the SP2, after being passed through a system of two Dekati DI-1000 ejector diluters, which diluted the aerosol in

pure nitrogen by a combined factor of 100. These were included to prevent saturation of the signals. The $CO_2$

concentration here was measured by a Li-Cor Biosciences (Lincoln, NE USA) model LI-6262 $CO_2$ analyser to

verify the level of dilution. Certified reference gases provided by BOC were used to calibrate all gas instruments.

The use of two $CO_2$ measurements in parallel (one on the flue and one on the nitrogen-diluted flow) was for the

purpose of evaluating the dilution factors of the ejector diluters. However, while both instruments were calibrated

using the same standards, the Li-Cor suffered from continuous drifts in its baseline, prohibiting a reliable real-time

measurement. The dilution factor could still be estimated by comparing the magnitude of abrupt changes within the

two measurements (e.g. onset of combustion). Because the dilutor flows were identical between experiments, the

dilution factor was assumed to be consistent and a value of 100 was used, based on an average of these comparisons.



An Aerodyne Compact Time-of-Flight Aerosol Mass Spectrometer (AMS) was used to measure the OA

concentration, as well as that of inorganic species including sulphates, nitrates and ammonium. This instrument has been documented extensively in the literature (see e.g. Drewnick et al., 2005; Canagaratna el al., 2007).

During this test series, the AMS was used in the fast mode (Kimmel et al., 2011), which allowed results to be collected with a much higher time resolution than the default 'MS' mode. The chopper alternated between 30 seconds in its open position, allowing particles to pass through, and 10 seconds in its closed position to establish the

background. This mode allowed a time resolution of 0.5 s during open phases. Particle size distributions were not measured as this requires the AMS to have a larger averaging time and use 'pToF' mode. Routine calibrations of the baseline, single ions and $m/z$ ratio were carried out before and after each test and where possible, nitrate ionisation efficiency calibrations were carried out between tests using particles of 350 nm, size-selected using a DMA.

As described by Allan et al. (2004), a fragmentation table was used to calculate the contribution of distinct chemical

species to each $m/z$ peak from raw AMS data. The abundance of nitrogen in this experiment due to the dilution required changes to be made to this table. Previous literature has remarked on the need to alter fragmentation tables for certain $m/z$ ratios when biomass burning is being measured (Ortega et al., 2013). The changes made for this experiment have been documented in Appendix 1. The collection efficiency for the AMS was assumed to be 1: this represents the probability that a given particle will stick to the vaporiser and be detected, so the mass concentration

must be divided by this to be quantitative. Although there has been no definitive assessment of the collection efficiency for biomass burning, previous studies in biomass-burning-dominant areas have used this figure (see e.g. Brito et al., 2014).

The SP2 was used to measure the concentration of refractory black carbon (rBC) in the emitted particles. rBC is carbonaceous, insoluble and vaporises only at temperatures near 4000 K (Petzold et al., 2013; Schwarz et al., 2008).

The operation and process of data interpretation have been described by Liu et al., (2010) and McMeeking et al., (2010). Particles pass through a 1064 nm laser beam inside the instrument and the consequent incandescent and scattered light is detected on a single-particle basis. The intensity of the laser is sufficient to vaporise absorbing material, which re-emits radiation in the visible spectrum on incandescence. Both scattering and incandescence signals are measured: the magnitude of the scattering signal is proportional to the optical diameter of a particle.

Scattering-only particles smaller than 200 nm cannot be detected. The incandescent signal is proportional to the mass of incandescent material, which is assumed to consist almost exclusively of rBC. Incandescent particles with a mass lower than 0.3 fg (approximately 70 nm diameter, assuming a density of 1.8 g cm-3) cannot be detected. The instrument was calibrated using Aquadag, with the standard correction of 0.75 applied (Laborde et al. 2012).

The SP2 has often been used to examine the thickness of organic or other non-incandescent materials that form

coatings on rBC particles (e.g. Liu et al. 2014; Liu et al. 2017). The presence of a coating enhances the scattering cross section of the particle, which allows the relative thickness of the coating to be determined using the methodology detailed in Taylor et al. (2015) and Liu et al. (2014). This approach was attempted here, but in this





case, rBC was often not emitted concurrently with organic material, which is likely to have resulted in extremely low coating thicknesses during these times. As the technique relies on the interpretation of only the leading edge to

reconstruct the scattering signal (i.e. as the particle enters the laser but before the coating begins to evaporate), very thin coatings on small particles are difficult to retrieve with this method, as the algorithm only utilises a very small number of data points with a very low signal-to-noise ratio to fit the 'tail' of a Gaussian distribution. Here, even the highest retrieval percentages were lower than 30%, which means that the successful fits were likely biased towards particles with larger coatings. As such, this data cannot be used.

The wood used in this experiment was rubberwood (*hevea brasiliensis*), a variety commonly used in West Africa as a domestic cooking fuel. This was sourced from Côte D'Ivoire. Each piece was cut from one of three similar lengths of wood with a diameter of approximately 10 cm, before being sanded down to ensure the highest possible consistency across the separate tests (see Figure ). The mass and dimensions of all samples were as uniform as possible and moisture content was approximately 7%. Four pieces of wood were used in each sample.


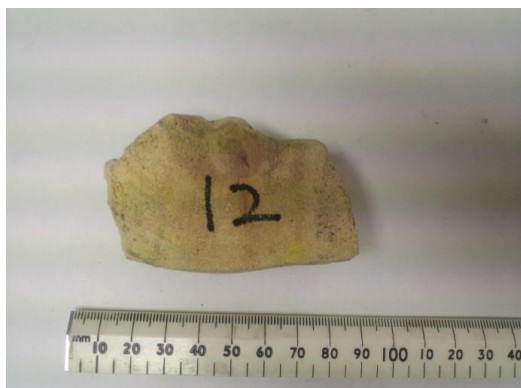

**Figure 2: An example of one of the wood samples used.**

The modified combustion efficiency (MCE) was calculated using the technique described by Akagi et al. (2011) and
Ward & Radke (1993) (Eq. 1):

$$MCE = \frac{\Delta CO_2}{\Delta CO_2 + \Delta CO} \qquad (1)$$

Where $\Delta CO_2$ and $\Delta CO$ refer to the $CO_2$ and CO enhancement respectively over background concentrations. This calculation yields the percentage of carbon released in the form of $CO_2$, assuming a small contribution from particulate matter. This assumes a negligible contribution from other gas phase organic compounds, including $CH_4$
and other, larger molecules. This can be constrained using results presented by McMeeking et al. (2009), who



showed that CO and CO2 together make up about 96% of carbon emitted during biomass burning. This method to calculate MCE is generally found to be accurate to within a few percent (Ferek et al., 1998).

The emission factors presented in Sect. 3.3 refer to the ratio of the mass of the emitted species to the mass of dry fuel consumed, in g kg$^{-1}$ (Cachier et al., 1995). The mass of the sample was measured during this experiment using a

mass balance in the FPA, allowing a more direct computation of this value than is often possible in field experiments during which the mass loss of fuels must be approximated from emissions of CO and $CO_2$. Emission factors were calculated as follows (Eq. 2):

$$EF = \frac{particulate\ emissions\ (g)}{sample\ mass\ loss\ (kg)} = \frac{\bar{X} \times f \times 10^{-1}}{\overline{\Delta M}} \qquad (2)$$

Where $\bar{X}$ is the average concentration of the species during the period being measured ($\mu$g m$^{-3}$), $f$ is the flow rate of

air in the exhaust tube (m$^3$ s$^{-1}$), $\overline{\Delta M}$ is the average mass loss rate of the sample (g s$^{-1}$).

Eight flammability experiments were undertaken, with different heat fluxes and flow rates. These, and the notation used to refer to them, are presented in table 1.

**Table 1: Outline of the burning environments used for the eight different tests carried out.**

| Burning environment | Heat = 30 kW m$^{-2}$<br>Flow rate = 200 lpm | Heat = 50 kW m$^{-2}$<br>Flow rate = 200 lpm | Heat = 50 kW m$^{-2}$<br>Flow rate = 50 lpm |
|---|---|---|---|
| Number of tests | 2 | 3 | 3 |
| Notation | *hF.1, hF.2* | *HF.1, HF.2, HF.3* | *Hf.1, Hf.2, Hf.3* |

Due to the emphasis placed on reproducibility during these tests, some aspects differ from real-world biomass burning and these must be taken into account when comparing these results with emissions reported elsewhere. The heating of samples throughout the experiments allowed greater reproducibility by reducing the effects of the external conditions, and as a result these data are more representative of burning of a larger accumulation of material. Emissions here are sampled in a duct rather than from an ambient environment and as such, there is no effect of

mixing with ambient air.

Each test began by initiating the air flow at the desired rate and instantaneously exposing the samples to the desired heat flux. A small flame was fixed above the sample, to drive piloted ignition of the pyrolysis gases when their concentration was sufficient. Ignition was defined to be the time at which a luminous flame was first visible. Similarly, extinction of each piece of wood was defined as the time when a luminous flame was no longer visible.

**3 Results**

Figure 3 shows the time series of aerosol and gas species being released during a test with 30 kW m$^{-2}$ heat flux and an air flow rate of 200 lpm (test *hF.1*). This experiment has been chosen for illustrative purposes as it demonstrates most of the different emission behaviours seen throughout these experiments. Ignition is indicated by the vertical line at t = 0 s and flame extinction by the period in dark grey (this begins with the extinction of the first piece of wood and ends with the extinction of the last). Between ignition and the extinction period, all four pieces of wood in the sample burned relatively steadily with a yellow-blue flame that decreased in intensity towards extinction. All four pieces of wood extinguished within 300 s of one another.


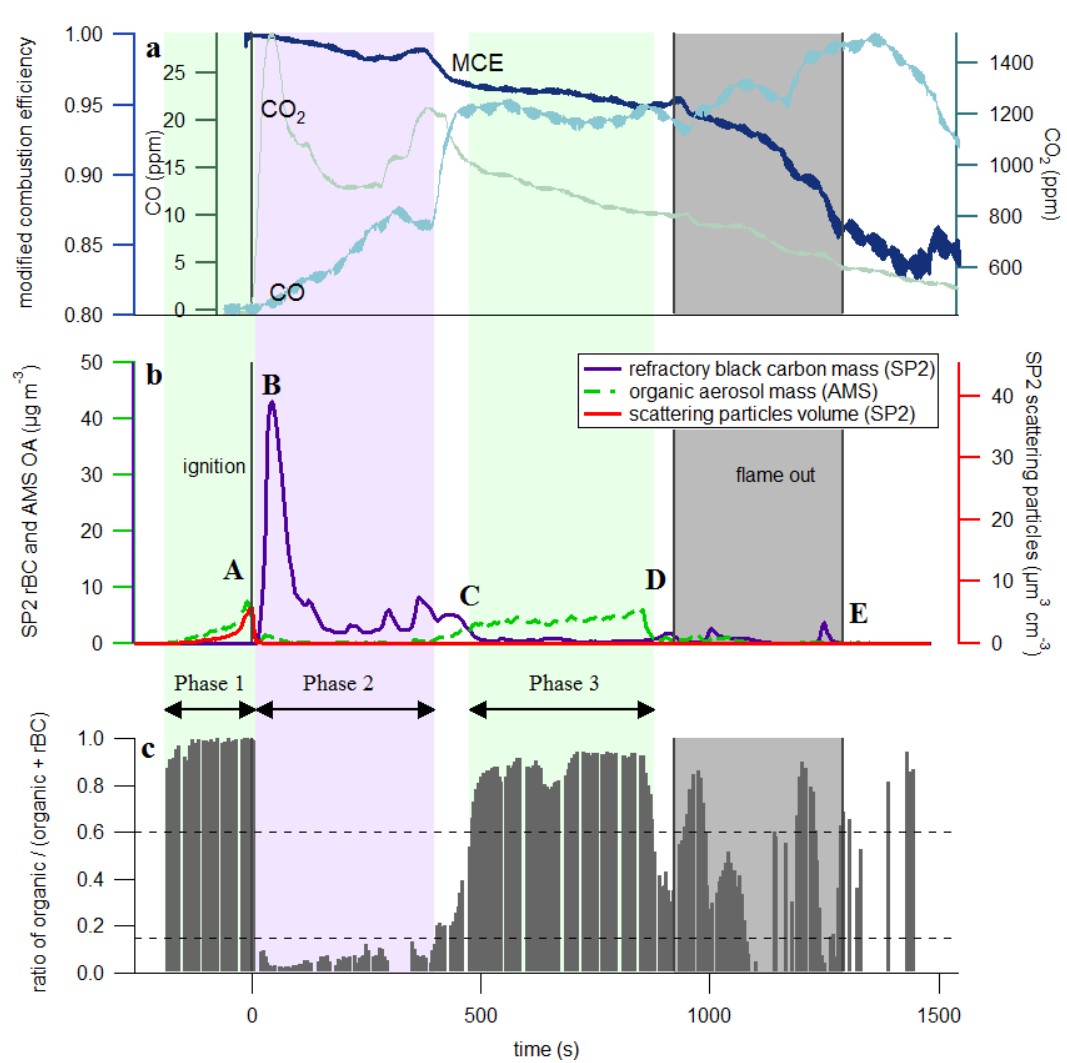

Figure 3: Three panels representing results as a function of time from a single combustion event under low heat (30 kW m$^{-3}$) and high flow (200 lpm) conditions. Panel a shows CO and CO$_2$ mixing ratios in green and blue respectively, with MCE in dark blue. Panel b shows the concentrations of particulate OA and rBC (mass measured in µg m$^{-3}$, not dilution-corrected) and particle volume measured by detected scattered light using SP2 (volume in µm$^3$ cm$^{-3}$). The mass derived from this volume can be obtained from the left hand y-axis, assuming a particle density of 1.1 g cm$^{-3}$. Panel c shows the ratio of OA to total aerosol. The green and red shading represent different combustion phases, established from the OA/total aerosol ratio. The grey area is the period during which the sample flames were extinguishing.

Throughout this experiment, noticeable changes were observed in the MCE, represented in panel *a*, which began to decrease gradually from unity after ignition. Combustion was highly efficient at the beginning of the experiment, as evidenced by the MCE remaining above 0.98 for the first 400 s, with a large quantity of CO$_2$ being produced. This is



typical for flaming combustion. As the experiment proceeded, the MCE decreased at a relatively steady rate until the first piece of wood extinguished, after which it dropped more rapidly from 0.95 to around 0.85. The general decline in MCE as the experiment progressed is typical of biomass combustion and can be seen in other laboratory experiments e.g. Heringa et al. (2012) during laboratory-based combustion in a log wood burner. This is typical of

the burning of charring solids in which the char layer results in a decrease in the pyrolysis rate and ultimately flame quenching (Bryden et al. 2002). After flaming combustion of the wood has quenched, any residual smouldering activity will produce relatively high yields of CO, which accounts for the sharp drop in efficiency during the extinction period (e.g. Lobert 1991; Urbanski 2013; Yokelson et al. 1997).

Figure 3*b* shows how measurements of rBC and OA varied over the course of the combustion event. Aerosol

measurements began as soon as the sample was exposed to the heat flux. A clear peak in OA could be seen prior to ignition as the sample was exposed to heat, which was also observed in the scattering volume measurement (**A** in Fig. 3*b*). This was likely due to the pyrolysis of the wood. During this process, gaseous pyrolysis products evolve which are volatile at high temperatures but are likely to condense into aerosol in the cooler environment of the exhaust. Almost no rBC particles were seen during this period, as a luminous flame is required for these to be

produced. The aerosol mass derived from the volume-convolved number distribution derived from SP2 scattering intensity signals are on the same scale as the AMS organic mass in Fig. 3*b* if a particle density of 1.1 g cm$^{-3}$ is assumed, which is consistent with organic biomass burning aerosol (Schkolnik et al., 2007). The similarity in magnitude seen in this image supports our assumption that the collection efficiency used for the AMS is unity.

Immediately after the flame ignition, there was an abrupt decrease in the production of OA, while rBC

simultaneously increased, peaking at just over 40 μg m$^{-3}$ (**B**). This large enhancement in the emission of rBC is associated with the onset of flaming combustion (see, for example, Bond et al., 2013; Kuhlbush & Crutzen, 1995). Flaming oxidises the pyrolysis gases to produce predominantly CO and $CO_2$. The MCE during this phase was high, remaining above 0.98. After reaching its peak 45 s after ignition, the concentration of rBC fell, although it continued to dominate particulate emissions. This accompanied a notable decrease in the $CO_2$ concentration (see Fig. 3*a*). At

this stage, char was observed on the top surface of the wood. The thermal properties of the char layer reduce the energy that is available to drive the pyrolysis reaction thereby decreasing the rate of production of flammable gases and decreasing the combustion intensity.

Approximately 450 s after ignition, the rBC mass reduced to become almost negligible and there was a large increase in organic mass measured by the AMS (**C**), which remained steady for 400 s. This was accompanied by a

substantial increase in the production of CO. This behaviour suggests a change from a flaming-dominant regime to one dominated by smouldering combustion, although a luminous flame was still visible. As the combustion became less intense and the flame retreated, oxygen was able to reach the surface of the sample, allowing smouldering combustion to occur alongside flaming. Although the MCE was considerably lower during this period than during the earlier, intense flaming combustion, it was still higher than is generally reported for smouldering combustion in

the literature (Akagi et al., 2011), although care should be exercised in the interpretation of this, as the definition of





'smouldering' is subjective and could be defined differently for different systems. The SP2 did not detect any light scattering particles at this time despite the comparatively large OA mass concentrations measured by the AMS. This suggests that the size distribution of organic particles here was significantly lower than during pyrolysis, with the majority of particles having an optical diameter smaller than 250 nm, the minimum size of non-incandescent particles detected by the SP2.

After 855 s, as the flaming combustion continued to decrease in intensity, there was a significant drop in the production of particulate matter (**D**). Flaming combustion of the whole sample had stopped by 1545 s, after which particle measurements were negligible (**E**). There was not an exclusively smouldering period that released particulate after the flames had extinguished, likely because the majority of the original mass of the samples had already been consumed by this stage.

Due to the high level of control and the small sample sizes used in this experiment, it has been possible to distinguish very clearly between these different periods of combustion. It is important to note that during real-world burning or larger-scale laboratory-based experiments, combustion will be more heterogeneous, with different phases of combustion highlighted here taking place simultaneously and the emissions free to form complex mixtures through mixing, condensation and coagulation.

Given the distinctive contrast in aerosol emissions at different times, as well as the recurrence of similar features observed during all tests, the ratio between the OA mass measured by the AMS and the sum of OA and rBC measured by the SP2, $r_{OA}$, was used to identify distinct phases of combustion ($r_{OA} = \frac{OA}{OA+rBC}$). In Fig. 3c, grey bars indicate the change in $r_{OA}$ throughout the burn: when this is high, OA dominates emissions; when low, rBC dominates. A ratio is not calculated when the total particulate emissions are less than 0.1 µg m$^{-3}$ due to noise in the signal.

As is illustrated in Fig. 3c, phase 1 represents pyrolysis immediately prior to ignition, where OA makes up more than 85% of the total particulate carbon emitted. Phase 2 is the period after ignition during which OA was less than 15% of particulate carbon produced (or rBC more than 85%). Phase 3 is smouldering-dominant combustion: the period during which organic mass contributed over 60% to total particulate carbon. As there is generally a swift transition from one regime to the next, the majority of the time series during combustion falls within one of these descriptions, although some tests, particularly those under *HF* conditions, contained more data in transition regions than others. The division described above allows the characteristics of each phase to be analysed separately and compared both within and between tests.

Results found in the literature generally support the assertion that OA dominates during pyrolysis and smouldering and more rBC is emitted during flaming than smouldering (e.g. Akagi et al. 2011; Kuhlbusch & Crutzen 1995; Simoneit et al. 1999). However, in the majority of experiments and ambient measurements, the proportion of OA emitted during flaming is significantly higher than the 15% used to identify the phase here. In a review of the



literature, Reid et al. (2005) report a summary of carbon apportionment from studies using thermal emission

techniques, with OA and black carbon reported as mass fractions. Most of the samples were taken from real-world

forest or Savanna fires. Using these values, it was possible to calculate a mean $r_{OA}$ of 0.84 for emissions reported to

be from flaming combustion, from 17 studies. The significant difference between this high value and the low one

reported in the present study is almost certainly due to the heterogeneous nature of real-world combustion compared

with the extremely controlled, high temperature and therefore more efficient combustion that takes place during

phase 2 here. It is also likely that during smouldering phases as described for more complex fires, weak flames are

still present between pieces of fuel, meaning that the definitions are inconsistent between these experiments. The

mean $r_{OA}$ of the three smouldering measurements was 0.95, which is consistent with phase 3.




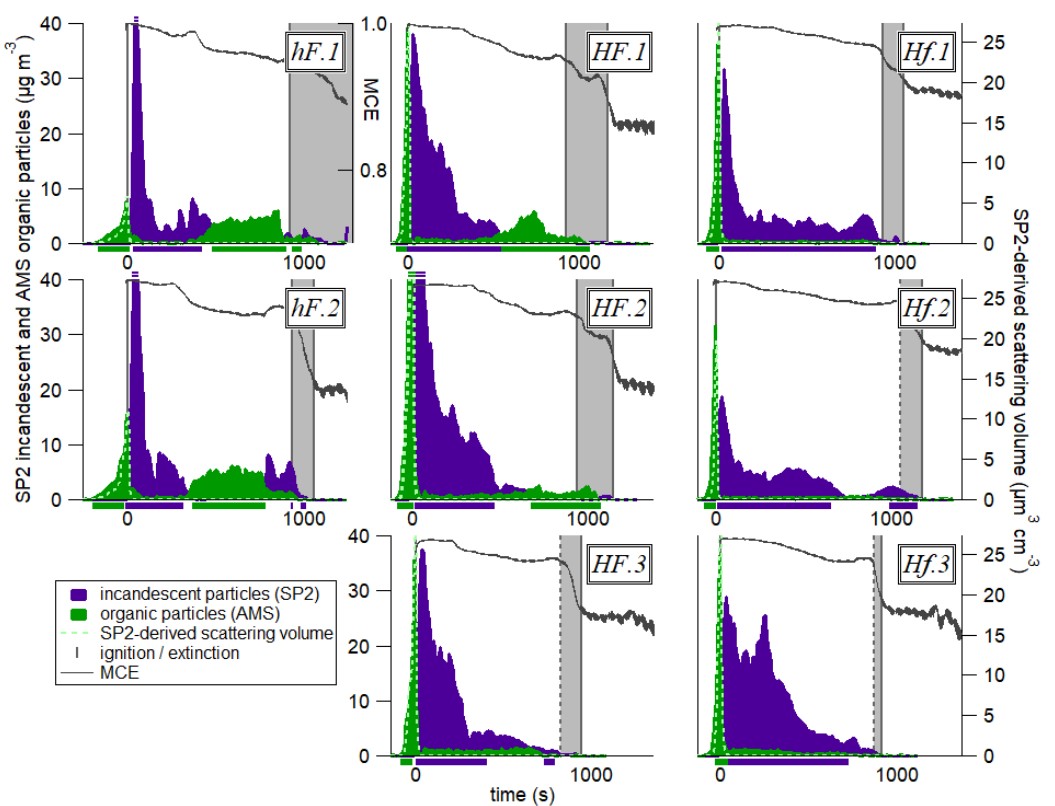

**Figure 4: Time series of the particulate properties during each of the eight tests carried out. The coloured traces show particulate concentration, as in panel b in Fig. 3, and the dark grey lines represent the MCE. The MCE scale on the first plot applies to all plots. The dark grey boxes show the time between the extinction of the first piece of wood and that of**

**the last. Where the first line is dotted, the first extinction was not recorded and the time is therefore an estimation based on the patterns seen across the tests in MCE and particulate concentrations around the time of extinction. Where two scored lines are shown at the concentration peak for a species, this means that the highest concentration for that species is above the scale for these plots; more information can be found in the supplementary material. The bars beneath each x-axis show which phase of combustion is taking place: phase 1 is the first green bar before ignition, phase 2 is the purple**

**bar and phase 3 is the green bar after ignition.**

Figure 4 shows the evolution of rBC and OA mass and SP2-derived scattering particle volume throughout each of

the eight combustion tests carried out. For each of these experiments, larger figures showing more detail, as in Fig.

3, are included in the supplementary material. The recurrence of the same phases described above can be seen in

these tests, with the coloured bar beneath each panel highlighting the phase of combustion taking place. The

difference in emissions from phase to phase is clearly much larger than the difference between overall emissions

released from different tests.



Nevertheless, differences between the different burning environments can be identified. For example, the peak in OA particle mass due to pyrolysis prior to ignition was noticeably smaller under low heat conditions than high. The influx of a large amount of energy from the infrared heaters under high heat conditions resulted in a faster and more intense period of pyrolysis before the flame ignited.

The shape of the rBC curve differed slightly between environments, with a far more gradual decrease under *HF* conditions after the initial peak. In general, the rBC under *Hf* conditions were the lowest, with a low peak followed by an abrupt decrease, although this is slightly less distinct in test *Hf.3*.

In both *hF* tests, substantial OA particle mass was seen later during the combustion period in phase 3, where it is likely that smouldering combustion was the dominant regime. Under *HF* conditions, there was sometimes a slight increase in organic mass loading later during combustion, but to a much lesser and less well-defined extent than in *hF*. OA-dominant behaviour was not seen at all after ignition in *Hf* tests. Due to the combination of a higher heat flux and a smaller cooling influence from incoming air in *Hf* tests, it is likely that the smouldering behaviour evident in the *hF* example described above was not able to become established. Instead, the high energy environment was able to maintain a more consistent luminous, rBC-producing flame throughout the entire period of combustion.

### 3.1 Organic aerosol mass spectra

Figure 5 shows the average mass spectra of organic particles from phases 1 (a) and 3 (b) as a percentage of the total OA released, along with the difference between the two phases (c). The mass spectrum from phase 2 is not considered here due to the very low concentrations of OA emitted. The two spectra shown are dominated by hydrocarbon ion fragments, including $C_nH_{2n-1}^+$ ($m/z = 41, 55, 69$), $C_nH_{2n+1}^+$ ($m/z = 29, 43, 57$), $C_nH_{2n-3}^+$ ($m/z = 67$, 81) and $C_{5+n}H_{5+2n}^+$ ($m/z = 77, 91$). These peaks are associated with fragments of saturated alkanes, alkenes, cycloalkanes and aromatic compounds, respectively. These are similar to spectra reported elsewhere during biomass combustion: for example, Schneider et al (2006) and Weimer et al. (2008) both observed organic spectra dominated by hydrocarbon ion fragments during laboratory-based biomass burning. The largest peaks seen in those experiments were generally *m/z* 43 or 44, and 29, as is also seen here. The peak at *m/z* 60 that is seen in both combustion phases here, though more predominantly in phase 3, is associated with fragments of levoglucosan and related anhydrous sugars. These are organic compounds formed from the pyrolysis of cellulose; *m/z* 60 is often seen together with smaller peaks at *m/z* 57 and 73 (Alfarra et al. 2007) and is used as a marker for biomass burning organic aerosol (BBOA), as so few other sources produce this peak (Simoneit et al. 1999). This is useful mainly for fresh aerosol, as the *m/z* 60 signal diminishes as the BBA ages (Cubison et al. 2011).

The peaks that were more prominent in phase 1 than phase 3 included the series *m/z* 41, 55, 67, 77 and 91. In phase 3, more prominent peaks were *m/z* 15, 29, 44, 57, 60 and 73. Heavier molecules were preferentially produced during phase 1. Thus, despite their similarities, it is clear that these two different processes produced distinguishable OA signatures.




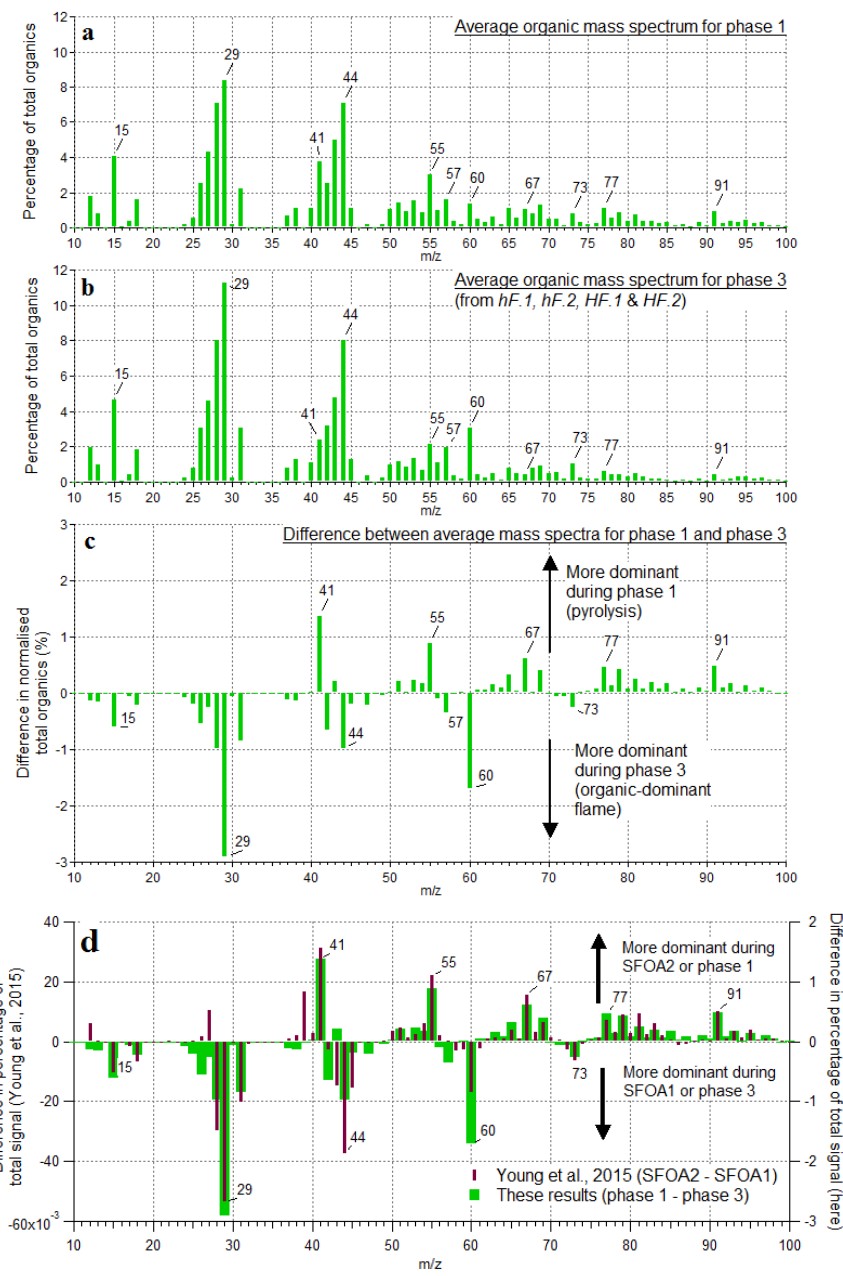

**Figure 5: The mass spectra of OA presented as a percentage of total OA produced during phases 1 (a) and 3 (b), and the differences between them (c). This is an average of the mass spectra for each phase across all tests from all burning environments. (d) shows a comparison between the difference of the two ambient solid fuel factors identified by PMF by Young et al. (2015) and the difference of the two organic phases observed here.**

355





An ambient investigation into the urban burning of solid fuels carried out in London in winter 2012 (Young et al., 2015) identified two different mass spectral factors contributing to the solid fuel organic aerosol (SFOA). These were found using positive matrix factorisation (PMF), a technique used to identify different components of organic aerosol contributing to the overall mix in ambient air (Zhang et al. 2011). The data were sampled from a large air mass with a number of contributing sources, of which two were identified to be SFOA. Various explanations for the origin of these two factors of SFOA were considered, with the authors concluding that different burning conditions was a likely reason that two different factors were produced. The difference between these two factors shows a remarkable similarity to the difference between the two combustion phases examined here; results from Young et al. (2015) are reproduced in Fig. 5d alongside the difference spectrum from the two organic phases observed in our controlled experiments.

Given the striking similarities between these two difference spectra, it is probable that the SFOA factors identified by Young et al. (2015) are related to the two OA-producing phases of combustion identified here, pyrolysis and smouldering. Young et al. (2015) noted that the peaks observed most strongly for the factor SFOA2, which are similar here to phase 1 (above the x-axis in Fig. 5d), are primarily composed of reduced hydrocarbons. Those in SFOA1, associated with phase 3 (below the x-axis in Fig. 5d), are composed of oxidised hydrocarbons. This is what would be expected from pyrolysis and smouldering aerosol respectively: before ignition, there is limited opportunity for aerosol to become oxidised, in contrast to aerosol produced while combustion is taking place.

A closer examination of Young et al (2015)'s results generally supports these associations. There was a significant time correlation between the two SFOA factors, which is consistent with the proposition that they were produced by the same sources during different phases of combustion. In some cases, SFOA2 (associated with pyrolysis) was seen to peak slightly earlier than SFOA1 (smouldering), which could be related to an increase in pyrolysis early during combustion. There was a slight dependency on wind speed and direction, with SFOA1 more likely to originate from the south and SFOA2 from the east or west. This cannot be immediately reconciled with the idea that factors are related to combustion phases, but could be related to different types of burning taking place in different areas.



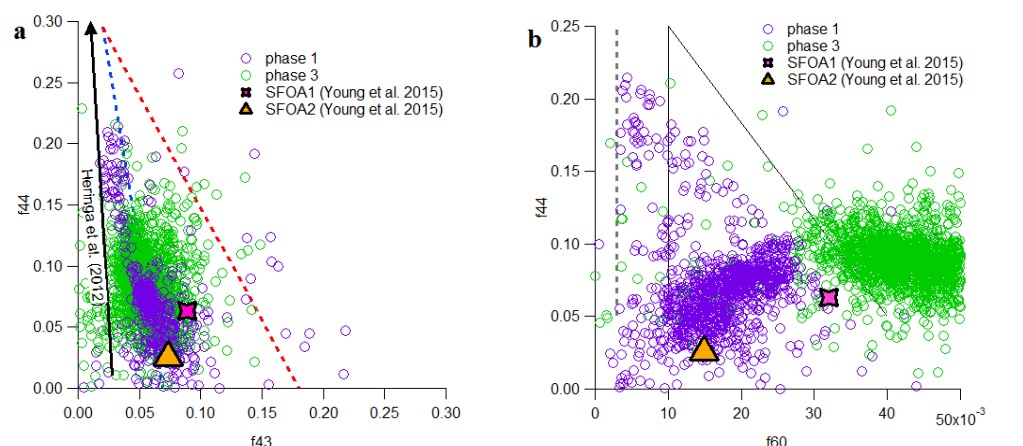

**Figure 6: f44 vs f43 (a) and f44 vs f60 (b), with phase 1 data shown in purple and phase 3 in green. The general trend of results from Heringa et al. (2012) is illustrated by the black arrow in (a) and results from SFOA2 and SFOA1 collected by Young et al. (2015) are reproduced in both plots.**

Comparing the relative contributions of significant *m/z* peaks shows further resemblances between the SFOA factors and our results. One example of this is the fractions of *m/z* 44 and 43 (f44 and f43), which are associated with the ions $CO_2^+$ and $C_2H_3O^+$ respectively. Although there is a contribution at f43 of $C_3H_7^+$, this lessens as the aerosol ages. These plots are often used to describe the ageing of OA in the atmosphere, with fresher aerosol populating the lower portion of the triangle shown in Fig. 6a and aged, more oxidised aerosol being found at the top. In general, ambient aerosol populates the inside of the triangle (Ng et al. 2010; Morgan et al., 2010). In contrast to ambient measurements, Heringa et al. (2012) found that fresh BBOA from laboratory-based log wood combustion contained lower f43. Thus, the majority of these datapoints fell outside the triangle to the left. On average, the f44 contribution increased as combustion developed.

In Fig. 6a, the data collected from phases 1 and 3 from all eight experiments are projected onto this space, alongside a diagrammatic representation of Heringa et al. (2012)'s fresh BBOA results and averages from Young et al. (2015)'s two SFOA factors. The data collected during this project are slightly closer to the ambient triangle than that of Heringa et al. (2012), but show a broadly similar pattern, with phase 1 data containing less f44 on average than phase 3, although there is considerable overlap between the phases. The same pattern is seen in the averages of Young et al. (2015)'s two SFOA factors: SFOA2 (phase 1, pyrolysis) contains less f44 than SFOA1 (phase 3, smouldering). SFOA1 contains more f43 than would be expected, but this is perhaps unsurprising, as f43 has been shown to be one of the most variable quantities in fresh biomass burning measurements (e.g. Crippa et al., 2014; Weimer et al., 2008).



A second comparison of *m/z* peaks, this time considering f44 vs f60, is shown in Fig. 6b. The peak at *m/z* 60 is associated with levoglucosan and other anhydrous sugars, which means it can be used as a marker for biomass burning. Similar to the first plot, ambient biomass burning aerosol is found within the marked triangle, with fresher aerosol at the bottom and more aged aerosol at the top. As the aerosol becomes more processed, f60 decreases. The

dotted line to the left shows the general location of ambient aerosol not associated with biomass burning (Cubison et al. 2011).

The two phases of the burn are separated horizontally in Fig. 6b. Phase 1 occupies the lowest, left-most part of the triangle, with low values of f44 and f60. This is generally within expected ambient boundaries for fresh biomass burning. Phase 3 OA, in contrast, is found to have higher f60 than is seen in ambient measurements. These very high

levels are likely due to the proximity of the measurements to the source, which allowed little opportunity for atmospheric processing. Data from the two ambient factors reported by Young et al. (2015), as shown in Fig. 6b, show the same pattern, with the SFOA2 data (associated above with phase 1, pyrolysis) sitting lower and to the left, and SFOA1 data (associated with phase 3, smouldering) sitting higher and to the right, although with a slightly lower average level of f60 than the very fresh aerosol that was produced during this experiment. Given that these

points do not lie on a line of negative correlation, Young et al. (2015) hypothesised that the difference between the two factors was due to more than just the degree of atmospheric processing, suggesting either fuel type, burn conditions or the phase of combustion as possible reasons for the difference. Data collected here strongly supports the notion that the phase of combustion producing them is the primary difference between these two different factors of SFOA.

The ratio of levoglucosan to potassium has been used in the past as an indicator of biomass combustion efficiency (Harrison et al. 2012). In high temperature fires, levoglucosan is more completely consumed by the flame and more potassium is emitted in particles, resulting in a low levoglucosan:potassium ratio from more efficient combustion. Young et al. (2015) found a lower ratio in SFOA1 (smouldering) than SFOA2 (pyrolysis), leading to the conclusion that the SFOA1 factor was from more efficient combustion.

However, Young et al. (2015) did not consider the possibility that a factor could be associated with pyrolysis rather than flaming or smouldering combustion. In our results, the lower levoglucosan during pyrolysis (judged from m/z peaks at 60, 57 and 73) cannot be ascribed to inefficient combustion because combustion, as such, was not yet taking place at this time. Furthermore, potassium was emitted almost exclusively during phase 2, so it was not related to either OA-producing phase. This could not be measured quantitatively by the AMS due to surface

ionisation (see e.g. Drewnick et al. 2006), but the timing of its qualitative emission was able to be established.

The clear similarities between Young et al. (2015)'s two SFOA factors and the mass spectra of pyrolysis and smouldering combustion presented here give a strong indication that our results could have an application in identifying ambient BBOA factors. Results presented here not only confirm that different phases contribute to BBOA variability by producing different mass spectra; further, they imply that it is possible to distinguish between



these two factors in ambient measurements. If future work is carried out to characterise the mass spectra produced by pyrolysis and smouldering combustion more completely, it may be possible to decrease the uncertainty in the BBOA factor constraints by assuming two related but independent biomass burning factors, rather than the current approach of using one.

Ambient studies have been carried out by both Zhou et al. (2017) in the western US and Brito et al. (2014) in South
America, which used PMF to establish three and two BBOA factors respectively. However, analysis of these factors in both cases suggested that the differences between them were due to the aerosol ageing in the atmosphere, rather than any difference in the phase of combustion at the source. Given that there was no ageing of the aerosol in our experiment, the features of phases 1 and 3 described here would likely both have contributed to the 'fresh BBOA' factor in each of these ambient studies. The relative remoteness of the sources and warmer temperatures for Zhou et
al. (2017) and Brito et al. (2014)'s measurements when compared with Young et al. (2015)'s likely contributed to the inability to distinguish between pyrolysis and smouldering emissions in these cases. This suggests that results presented here are likely to be most valuable in situations where measurements are made in close proximity to the biomass burning source and where atmospheric processing of aerosol is limited.

### 3.2  Nitrogen

Previous experiments have found that the proportion of organic to inorganic material in aerosol can affect its hygroscopicity, with organic aerosol tending to be hydrophobic and inorganic material being more likely to take up water (see e.g. Hersey et al., 2011; Shinozuka et al., 2009). This can therefore affect a particle's likelihood of becoming a cloud condensation nucleus and change its scattering characteristics. Furthermore, nitrated phenols (R-$NO_2$), which have been observed previously in biomass burning emissions, have been shown to absorb light at short
wavelengths (Mohr et al., 2013; Zhang et al., 2016). This will have an impact on the radiative effects of BBOA. Organic nitrates (R-O-$NO_2$) have a present, though smaller radiative effect (Roberts and Fajer, 1989), but can affect $O_3$ production by acting as a reservoir for NOx (e.g. Perring et al. 2013). Given these atmospheric impacts, it is valuable to establish how much nitrogen emitted from biomass burning is organic and how much is inorganic in nature.

Farmer et al. (2010) established a method by which to determine the proportion of inorganic nitrate to the total using the ratio of $m/z$ 46 to $m/z$ 30 measured by the AMS, which correspond to the $NO_2^+$ and $NO^+$ ions, respectively. It is assumed here that the $m/z$ 30 signal measures exclusively ammonium nitrate and oxidised nitrogen-containing organics. Although interference from mineral nitrates (such as $KNO_3$ or $NaNO_3$) is possible, it is unlikely to play a significant role here: potassium was released during a different phase of the burn to nitrate, and previous studies
have shown that such mineral nitrates are not measured well by the AMS due to their size and vapourisation efficiencies (see e.g. Reyes-Villegas et al., 2017). Although interference from $CH_2O^+$ is possible, it is unlikely to be a major factor as its concentrations are typically low compared with other organic peaks (Reyes-Villegas et al. 2017).





Ratios of *m/z* 46:30 have been shown to be higher for inorganic nitrate than for organic nitrate, as ammonium nitrate

fragments more strongly to *m/z* 46. For example, Alfarra et al. (2006) and Fry et al. (2009) found *m/z* 46:30 ratios

between 0.33 and 0.5 for ammonium nitrate. Hao et al. (2014) and Sato et al. (2010), in contrast, measured ratios

between 0.08 and 0.36 for particulate organic nitrates. Reyes-Villegas et al. (2017) confirmed the presence of

nitrogen-containing organics during bonfire night 2014 in Manchester, UK, from a measured ratio of 0.11-0.18.

Though the ratio was not calculated in their study, Bahreini et al. (2005) saw considerably higher *m/z* 30 that 46

signals, which was attributed to nitro-organics. For some experiments here, it was not possible to establish the *m/z*

46:30 ratios due to low emissions of these factors during combustion, particularly for experiments that did not

demonstrate phase 3 behaviour. Using data from experiments *hF*.1, *hF*.2 and *HF*.1, a median *m/z* 46:30 ratio of

$0.084 \pm 0.15$ was found, which did not change significantly between phases and sits within the range for organic

nitrate found in the literature (Hao et al., 2014; Sato et al., 2010; Reyes-Villegas et al., 2017). $NO^+$ and $NO_2^+$ were

emitted concurrently with OA throughout with a ratio of 0.29:1 and a correlation of $r2 = 0.99$. These results strongly

suggest that the nitrogen emitted during this experiment was organic.

### 3.3  Emission factors

In atmospheric models, the emissions from wildfires, cookstoves and other sources of BBA are often represented by

average emission factors (EFs), which relate the production of a given pollutant to the mass of fuel that has been

burned (e.g. Liousse et al. 2010). The average EFs employed by these models are measured in the field or in

laboratories for specific fuels and, occasionally, burning conditions. They are compiled into inventories, such as

those presented by Andreae & Merlet (2001) and Akagi et al. (2011). However, very large variation in EFs can be

found even within individual datasets; uncertainties are attributed to fuel type and characteristics, combustion

phases, fire intensity and the naturally chaotic behaviour of fires (Reid et al. 2005). Results presented here provide

some insight into some of the mechanisms that result in this high variability.

In order to compare results collected here with those in the literature, average EFs for the duration of each phase of

combustion have been calculated for OA and rBC for each of the experiments. These, alongside average MCEs for

the entire combustion period, are displayed in table 2. The average OA and rBC emission factors for each entire

burn are plotted against overall MCEs in Fig. 7.

**Table 2: Emission factors for OA and rBC during each phase for each test.**

| Test | Phase 1 OA (g kg⁻¹) | Phase 1 rBC (g kg⁻¹) | Phase 2 OA | Phase 2 rBC | Phase 3 OA | Phase 3 rBC | MCE |
|------|------|------|------|------|------|------|------|
| *hF.1* | $1.20 \pm 0.07$ | $0.0163 \pm 0.0007$ | $0.028 \pm 0.002$ | $0.81 \pm 0.04$ | $0.61 \pm 0.01$ | $0.088 \pm 0.002$ | 0.958 |
| *hF.2* | $2.0 \pm 0.1$ | $0.019 \pm 0.001$ | $0.044 \pm 0.002$ | $1.09 \pm 0.06$ | $0.70 \pm 0.01$ | $0.053 \pm 0.002$ | 0.964 |
| *HF.1* | $4.5 \pm 0.7$ | $0.0075 \pm 0.0009$ | $0.044 \pm 0.009$ | $1.13 \pm 0.03$ | $0.48 \pm 0.01$ | $0.096 \pm 0.002$ | 0.967 |
| *HF.2* | $3.4 \pm 0.4$ | $0.04 \pm 0.01$ | $0.053 \pm 0.003$ | $1.31 \pm 0.03$ | $0.312 \pm 0.007$ | $0.124 \pm 0.003$ | 0.967 |
| *HF.3* | $4.2 \pm 0.4$ | $0.012 \pm 0.001$ | $0.053 \pm 0.001$ | $1.19 \pm 0.02$ | | | 0.973 |
| *Hf.1* | $1.9 \pm 0.2$ | $0.011 \pm 0.002$ | $0.0268 \pm 0.0007$ | $0.48 \pm 0.01$ | | | 0.981 |



| | | | | | |
|---|---|---|---|---|---|
| *Hf.2* | 2.8 ± 0.4 | 0.0042 ± 0.0006 | 0.0268 ± 0.0007 | 0.459 ± 0.008 | 0.977 |
| *Hf.3* | 1.5 ± 0.2 | 0.009 ± 0.002 | 0.0581 ± 0.0007 | 1.00 ± 0.01 | 0.981 |

Within each experiment type and phase shown in table 2, the EFs for each species were reasonably consistent; the variation from the mean in 13 cases out of 16 was less than 25%. It is important to note, however, that these sample sizes were extremely small (n = 2 or 3). A large number of studies have suggested that the phase of combustion (flaming or smouldering) is the best indicator of particulate emissions. For example, McMeeking et al. (2009) found a strong correlation between combustion phases and emission factors, but only a weak correlation with fuel type. A similar result was noted by Weimer et al. (2008), who observed a distinct difference between particulate emissions from different phases, but almost none between wood types. This is supported here: the greatest differences in particle emissions were between individual phases of combustion, rather than being between different burning environments.

Considering the entire combustion period instead of individual phases, Fig. 7a shows a negative correlation between the overall OA emission factor and MCE, with both factors being strongly influenced by the burning environment in which combustion took place. Low heat and high flow (*hF*) tests yielded the lowest MCE and highest OA EFs, while high heat and low flow (*Hf*) resulted in a high MCE and low OA EFs. This is largely in line with what would be expected: less efficient combustion is generally understood to result in OA being released in higher quantities. This was closely related to the presence and duration of phase 3. The four experiments with the lowest OA EFs are, as would be expected, the four in which there was no OA-producing phase 3.

The trend in emission factors for rBC is less clear. Previous literature suggests that a positive correlation can be expected between rBC emissions and MCE (see e.g. Andreae & Merlet, 2001; Christian et al., 2003; McMeeking et al., 2009), due to the preferential emission of rBC during more efficient flaming combustion. Such a correlation could be seen here for tests at a high flow rate (*hF* and *HF*), but only one of the three low flow cases continued this trend, with the other two having much lower EFs.





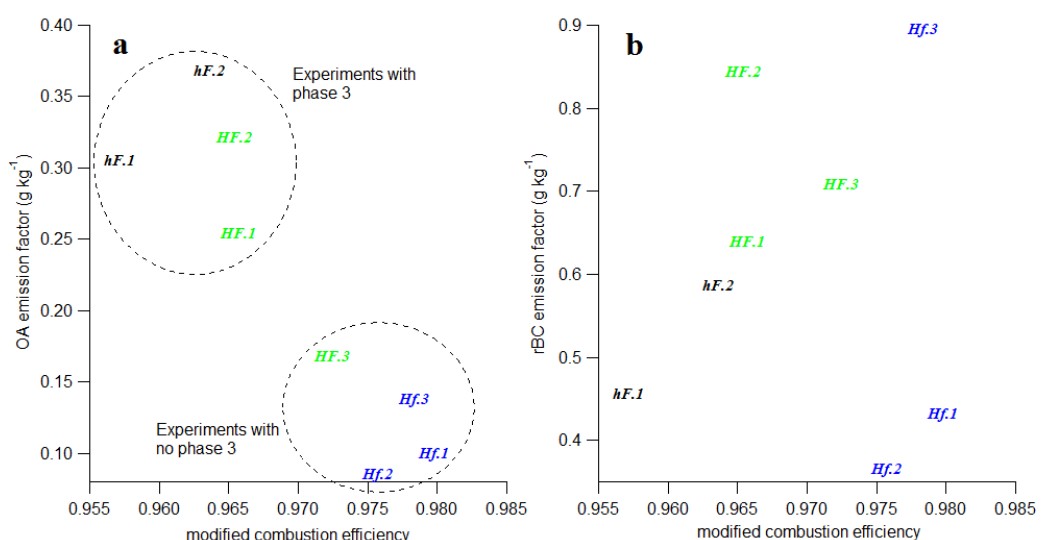

**Figure 7: overall OA and rBC emission factors for each test.**

Andreae & Merlet (2001) reported average EFs for biofuel burning under ambient conditions of 4.0 g kg$^{-1}$ and 0.59 g kg$^{-1}$ for organic carbon and black carbon respectively (based on a summary of data from a number of different studies). Michel et al. (2005) used values between 2.1 and 6.4 g kg$^{-1}$ to represent OA EFs and between 0.6 and 0.725

g kg$^{-1}$ for black carbon. For both of these examples, BC EFs are comparable with ours, but OA EFs are an order of magnitude higher. This is likely explained by the high efficiency of combustion in our experiments compared with that of real-world combustion, and this discrepancy is resolved when experiments at similar MCEs are considered. In a series of laboratory-based experiments, McMeeking et al. (2009) observed a strong correlation between OA EFs and MCE, with emission factors ranging from around 0.5 g C kg$^{-1}$ at high MCE to 50 g C kg$^{-1}$ for low MCE; this

correlation, as with ours, was attributed to the increasing amount of smouldering combustion at low MCE. Results presented here are comparable with McMeeking et al. (2009)'s high MCE results and show a similarly clear negative trend. The same study saw a less obvious correlation between elemental carbon EFs (which can be compared with rBC here) and MCE. This was particularly variable above an MCE of 0.95. Thus, the variability of rBC EF results collected here fits well with results in the literature.




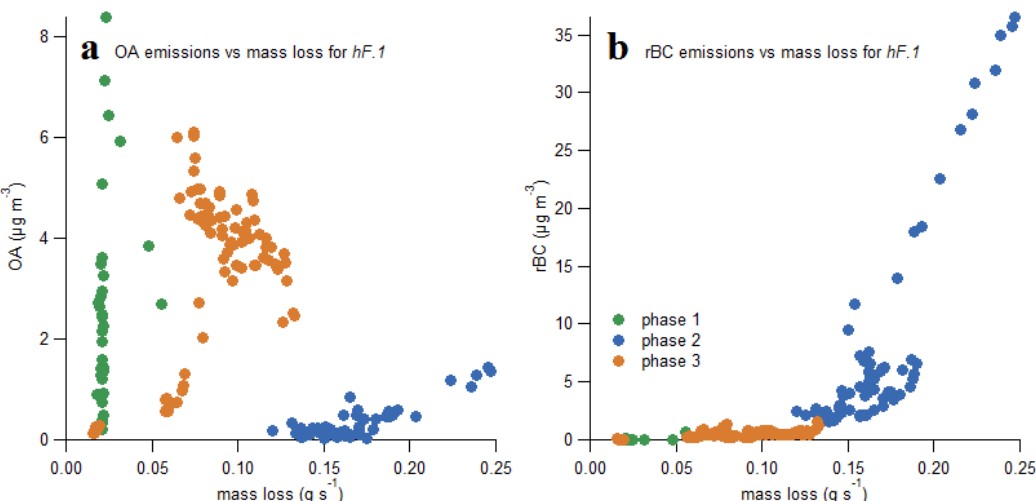

**Figure 8: OA (a) and rBC (b) concentrations against mass loss, averaged over every 5 seconds.**

Due to the high time resolution of the experiments carried out here, it was possible to examine the change in EFs in

545 more detail throughout each phase. The change in rBC and OA concentrations with the rate of mass loss for

experiment *hF.1* is presented in Fig. 8, which illustrates key features that recur throughout the test series.

There was little correlation between emissions of OA and the mass loss of the sample during either phase 1 or 3, the

two phases when the vast majority of OA was emitted. This illustrates the highly variable nature of smouldering

combustion, even within these well-controlled experiments. Very low rates of mass loss during pyrolysis yielded

550 extremely high and variable OA emissions, which suggests that mass loss during this phase is a particularly poor

indicator of expected emissions. These results show that the most reliable indicator of the OA emitted from

combustion is the efficiency of combustion and the duration of phase 3.

The rBC emissions always increased with increasing mass loss. An interesting feature is that above a threshold rate

of mass loss in each experiment, the rate of rBC emission substantially increased. In Fig. 8 this transition can be

555 seen at a rate of mass loss of approximately 0.15 g s$^{-1}$, after which the gradient becomes noticeably steeper. Thus, if

a flame reached a high intensity and rate of mass loss sooner, it produced disproportionately more rBC. This feature

could explain why there is more variability in the emission of BC at high MCEs in studies such as that carried out by

McMeeking et al. (2009): above this threshold, small changes in the intensity will have a much larger effect on rBC

emissions

560 The presentation of three phases here, rather than the standard two, provides some insight into the large variation in

EFs reported in the literature and the mechanisms producing these. EFs are typically examined in relation to two

variables: the fuel mass loss and the MCE, which is used as a proxy for the proportion of flaming or smouldering



combustion. Nevertheless, studies have shown that less than 50% of the variance in particle EFs can be accounted for by the MCE (Ferek et al. 1998; Janhäll et al. 2010; Reid et al. 2005). Comparable uncertainties are seen in
relating emissions directly to mass loss (e.g. Freeborn et al. 2008; McMeeking et al. 2009).

Pyrolysis is a process of thermal degradation rather than of combustion, so it cannot be easily parameterised using these two combustion-related variables. Nevertheless, both the large concentrations of OA released during phase 1 here, and with the comparison with Young et al. (2015)'s ambient results shown in Sect. 3.1, suggest that direct pyrolysis emissions could account for a significant proportion of ambient BBOA aerosol described in the literature.
It is therefore possible that the current approach of describing biomass burning emissions using mass loss and MCE as key variables is unable to account for emissions released during this third phase. This could begin to explain some of the large variability in biomass burning EF measurements, even between tests with similar MCEs and within the same investigation. If this is the case, identifying a further parameter that is able to capture pyrolysis OA could add a significant further constraint to bottom-up studies of EFs.

**4 Summary and conclusion**

Biomass burning experiments were performed under highly controlled test conditions in a Fire Propagation Apparatus. Eight experiments were performed in three different burning environments. In each case, the samples were exposed to heat fluxes of 30 or 50 kW m$^{-2}$ and an imposed airflow of 50 or 200 lpm of air. Emissions of organic aerosol and refractory black carbon were measured in real time using an Aerosol Mass Spectrometer and a
Single Particle Soot Photometer, respectively.

Emissions were observed to be dependent on three separate phases during small-scale wood burning: pyrolysis (phase 1), which produces mostly OA; flaming combustion (phase 2), when most of the rBC is produced; and smouldering-dominated combustion (phase 3), which emits primarily OA, although a luminous flame can still be seen. The emissions from any real-world burning event will comprise a variable combination of these phases taking
place simultaneously within a mass of burning material. We have shown that the relative abundances of key gas and particulate components emitted during each particular combustion phase are similar across tests in different burning environments. However, the environment in which combustion takes place affects the duration of each phase, and smouldering behaviour is only observed when conditions exist to lower the flaming intensity.

Our observations show that the particulate emitted during pyrolysis and smouldering is primarily OA, and that the
OA in each of these burn phases is chemically different. The composition of the former contains more reduced hydrocarbons and the latter is characteristic of more oxidised hydrocarbons. The ratio of f44 to f60 as measured using an Aerosol Mass Spectromer differs between these two phases, with smouldering having a much higher proportion of f60. This can be distinguished from the process of aerosol ageing, as aged aerosol would show lower quantities of f60 alongside greater quantities of f44, which is not seen here.





The differential chemical characteristics of these two phases are similar to those of two solid fuel organic aerosol factors identified by Young et al. (2015) in ambient air. This suggests that the differences in burn phases can contribute to the variability of ambient BBOA mass spectra and confirms that it is possible to infer burn conditions based on measurements of fresh emissions.

The emission factors for both aerosol species were consistent with results in the literature from experiments with similar modified combustion efficiencies. Within each experiment type and phase, emissions were reasonably consistent. Averaging over the entire period of combustion showed better correspondance between EFs and the MCE for OA than for rBC, which is consistent with results from previous experiments. However, the release of OA during pyrolysis and smouldering were not coupled to mass loss, which suggests that the correlation with MCE is best explained by the duration of the smouldering phase. The variability in rBC EFs could be related to a large

increase in the rate of rBC emission above a certain threshold flame intensity.

    The presentation of different chemical characteristics for three phases of biomass burning here provides a physical basis that could help to explain why there is such large variability in biomass burning emissions. This variability creates difficulties when producing bottom-up representations of emissions. If the difference in emissions associated with the pyrolysis and smouldering phases are better understood, however, and if further approaches to quantifying

pyrolysis are developed, it could be possible to constrain emissions estimates more completely.

    By being able to reproduce fire characteristics consistently between experiments and control the combustion environment we have been able to explore and separate processes that form different aerosol components during its evolution. In doing so we have been able to relate a number of key aspects of the burn to real world systems and to highlight why the systems behave in the way they do. Our work shows that whilst black carbon emission factors can

be used robustly caution needs to be used when applying OA emission factors averaged empirically over whole burns to other systems since the emission of OA is decoupled from the total mass loss. Further exploration of aerosol yields from such carefully controlled fires and relation to larger-scale experiments may help to deliver greater constraint on variability of particulate emissions in atmospheric systems.

    *Data availability.* Raw and processed data are archived at the University of Manchester and available on request.

*Author contributions.* S.L.H., J.C.T., R.H., J.D.A. and H.C. designed the project; S.L.H., J.C.T., W.T.M., J.D.A, P.I.W. and K.S. operated, calibrated and performed QA of instrument measurements; all authors contributed to the interpretation of data; S.L.H led the manuscript preparation, with J.D.A., H.C., J.C.T., R.H., D.L., K.S. and C.L contributing and critically reviewing.

    *Acknowledgements.* This work was supported by the UK Natural Environment Research Council (NERC) through
the SAMBBA project (grant ref: NE/J010073/1) and the lead author was supported by the NERC Doctoral Training Programme (grant ref: NE/L002469/1).



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
