# Peer review of "Highly-controlled, reproducible measurements of aerosol emissions from combustion of a common African biofuel source"

_Atmospheric Chemistry and Physics, 2017_

## Referee Comment (RC1) · Anonymous Referee #1 · 8 Sep 2017

This work studied aerosol emissions from burning of wood samples from west Africa in a highly-controlled manner. Parameters that were regulated include the shape and mass of wood samples, airflow and surrounding thermal environment. Measurements of OC and rBC emissions were performed online using a compact-AMS and an SP-2 respectively after $\sim$ 100-time dilution. CO and CO2 concentrations were also measured, thus modified combustion efficiency – a qualitative index for combustion condition – was determined. The highly controlled burns coupled with fast real-time measurements made it possible to characterize the fast changes in aerosol emissions over the course of combustion and relate the changes directly to combustion conditions. The connection of the results of this study and those from an ambient study by Young

et al. is rather interesting and provides an example for using the results of this study to interpret ambient BBOA factors and differentiate BBOAs emitted from different burning conditions. This is a quality work that generated new results on emissions and characteristics of aerosol from burning and pyrolysis of wood. The manuscript is overall well written and the scope of the work fits well within that of ACP. I recommend acceptance after the authors respond to the following comments.

Speaking of the repeatability of combustion events, which is the core of this work, one question is whether there is quantitative information to demonstrate this. For example, were the same experimental conditions repeated and how the emission rates and aerosol characteristics varied for the same condition? How do the emission factors determined in this study compare with the values reported in literature?

It was mentioned at the end of page 3 that the usage of a pilot flame had a negligible influence on CO and CO2 emissions. Are there measurements data to demonstrate?

Line 164, how was fuel moisture content determined?

In the calculation of EF in this study, the loss mass was determined by weighing. But how was the effect of the dryness of the fuel accounted for? It would be interesting to also report EFs based on total carbon burned/emitted, e.g., normalized by total carbon measured in CO and CO2. This may allow more direct comparison with field observations.

Since dilution may change the concentration profile, in Fig. 3, it might be interesting to add the 2nd CO2 measurement data on panel b.

What's the detection limit of SP2 for rBC? How much of the org/(org+rBC) variation under low rBC conditions shown in Fig. 3c was due to noise in the rBC measurement?

Line 291, it would be interesting to provide the range of rOA for flaming combustion or quote the "high" and "low" values (and citations) for the subsequent sentence.

Figure 4 is a somewhat difficult to read, I suggest adding an MCE axis in the 2nd and

3 row. Adding ticks on the MCE axis may also be helpful.

With regard to the paragraph underneath Figure 6: for BBOA, f44 and f43 are not necessarily primarily associated with CO2+ and C2H3O+ respectively. There can be considerable C2H4O+ and C3H7+ in the MS of BBOA.

Bottom paragraph on page 19, although CH2O+ is typically low compared to other organic peaks, since Org/inorg ratio is high in BB aerosol, it can nevertheless be an important contributor to the signal at m/z 30. Organic contribution to m/z 46 can be even more important. These issues should be more clearly discussed since the c-AMS used in this work can't differentiate ions with the same nominal m/z. Also, what was the 46:30 ratio for ammonium nitrate measured during this study?

Line 505 – 508, in addition to the two studies mentioned here, a recent study by Collier et al. (ES&T, 50, 8613–8622, 10.1021/acs.est.6b01617, 2016) also reported a negative correlation between OA emission factors and MCE for wildfires.

---

## Referee Comment (RC2) · Anonymous Referee #2 · 12 Sep 2017

Review of Manuscript acp-2017-679 Title: Highly-controlled, reproducible measurements of aerosol emissions from African biomass combustion Authors: Sophie L. Haslett, J. Chris Thomas, William T. Morgan, Rory Hadden, Dantong Liu, James D. Allan, Paul I. Williams, Keïta Sekou, Cathy Liousse and Hugh Coe

Overview: I find this manuscript to be well written and logically organized. The manuscript describes 8 biomass burning measurements of organic aerosol (OA) and refractory black carbon (rBC) from one type of rubberwood material in a highly controlled combustion system. The combustion system was designed through international standards for measuring combustion properties of different fuels for industrial

and insurance purposes. This is (at least) one of the first studies that start to bridge the gap between industrial standard combustion practices and ambient environmental studies. I applaud the authors for taking this approach. As these highly detailed measurements are important and there are few such measurements reported in current literature, this manuscript is both timely and appropriate material for ACP. The manuscript should be published with attention paid to the following minor issues.

General comments:

1.) ". . .from African biomass combustion" in the title implies a broader appeal than the actual use of 32 pieces from one specific type of rubberwood tree. Perhaps, ". . . from combustion of a common African biofuel source".

2.) What was the mass of the combusted fuel? The only details given are one dimension (100 mm), a picture, and ~7% moisture value.

3.) Interestingly, the authors do not emphasize the use of an industrial standard equipment for controlling the combustion process. This is a ready mechanism for others in the field to conduct similar experiments, such that a bit more detail might help move the whole field further along.

4.) In the discussion of the OA mass spectra, is there a reason why PMF was not done on the current data set to see how well it reproduces the average spectra for the two conditions (phase 1 and 3)?

5.) In the discussion of Figure 6, placing either PMF factors or phase 1/3 weighted means on the plots would help in interpreting the discussions of the relative abundances of f44 vs f43 and f44 vs f60. For example, line 395 "On average, the f44 contribution increased as combustion developed" is stated without an average value on the plot to assess. Attempting to discern "averages" of data points on a plot is difficult for readers, especially when the markers overlap.

6.) Differences in the PMF factors from Young et al. showed very nice correlations

with the differences in the mass spectra from phase 1 and 3. How do the actual PMF factors compare with the phase 1/3 mass spectra? While the differences are important, the actual connection in the text is that the Young et al. SFOA1 and SFOA2 may well represent phases 1/3 in ambient, which is a comparison of the absolute individual spectra, rather than just differences.

7.) During the discussion of ambient measurements (around lines 505 and 560+), the authors cite Zhou et al (2017) and Brito et al (2014), but appear to have missed Collier et al (2016), which showed a very strong correlation between OA/(CO2+CO) (which is a type of ER or emission ratio, rather than EF or emission factor, given the lack of information on fuel mass) and MCE in ambient wildland fires in the western US. The current lab results and these ambient results are in strong agreement about how OA emissions are governed in large part by combustion conditions for similar fuel types. They may differ by the pyrolysis emissions and/or effects of heat. The implications emphasized herein and noted in Collier et al. are important from both ambient and lab viewpoints and could readily be included.

8.) Lines 483-485. The authors appear to give a median values of m/z 46:30 as 0.084+/= 0.15 and linear correlation with a slope of 0.29. Are these values from the same distributions, suggesting that the m/z 46:30 distribution is highly skewed? What does a skewed distribution say about the results?

9.) The authors rightly emphasize the potential impact of pyrolysis OA emissions in the underlying emission factor variations observed with MCE (especially lab studies); however, the authors thereby downplay differences in heat load on fuels for a given fire. Here, the authors include this parameter explicitly (perhaps for the first time for ambient-related measurements). The heat load can vary depending upon whether the combustion occurs inside a stove (i.e., solid fuel combustion) or open air (wild land combustion). It can also vary by how much fuel is available (both in lab and ambient) and how well packed. Might this have a large an impact on measured OA emissions per MCE, similar to that suggested for pyrolysis?

---

## Referee Comment (RC3) · Anonymous Referee #3 · 16 Sep 2017

This paper reports organic aerosol (OA) and refractory black carbon (rBC) emissions along with gas phase carbon dioxide ($CO_2$) and carbon monoxide (CO) emissions from controlled laboratory experiments where the same amount and type of fuel was burned with varying applied heat and flow rates of air through the sample. The results are relevant to the rapidly growing body of work on aerosol emissions from biomass burning. A major finding of this work is that there are distinct types of aerosols from each of the three different phases of the controlled fires: predominantly less-oxidized OA during the initial pyrolysis phase (Phase 1, prior to ignition with no emissions of $CO_2$ or CO), almost entirely rBC during the flaming phase (Phase 2, after ignition with a relatively high modified combustion efficiency or MCE), and a mixture of mostly more-

oxidized OA with a small amount of rBC during the smoldering phase (Phase 3 after ignition with a relatively low MCE). The relative amounts of aerosols between the three phases were generally consistent between experiments with the same heat+flowrate conditions. The results of these laboratory experiments were compared with other laboratory and field measurements.

The manuscript is fairly clear to understand, but some of the analysis and conclusions are confusing given the types of experiments conducted. This paper is important to many readers of Atmospheric Chemistry and Physics and should be published after some clarifications are made to the manuscript. The key areas needing revisions are described below.

Overall Comments:

1) The manuscript is lacking a conceptual picture of the processes examined in these experiments, and interpreting the results depends on such a foundation. For this study, the heat applied to the sample (high heat = H or low heat = h) and the flow rate across the sample (high flow = F or low flow = f) were varied for 8 experiments. How should the results vary with these changing experimental conditions? It is likely dependent on the applied heat and flow rate in different ways for the different phases.

2) Prior to ignition, it is expected that the amount of heat applied and sample flow rate will affect the amount of vapors up to their flashpoint (ignition) as well as the amount available for subsequent re-condensation as OA. Higher heat applied increases the vaporization rate of semi-volatile species from the sample. Higher/lower flow rate would then decrease/increase the concentration just prior to ignition. rBC is not expected to be formed during the pyrolysis phase since there is no combustion occurring (no $CO_2$ or CO). What would happen if an ignition source was not there? Since this is an important phase for producing OA and is not a combustion phase, how might the fuel type be relevant for the emissions of this phase? What kind of differences might be expected between wood used as cooking or heating fuel versus wildfires? A little more

discussion on this subject should be included in the paper.

3) After ignition, how does the applied heat compare with the heat produced by combustion for the flaming and smoldering phases? Applied heat may be irrelevant to the results if the combustion processes generate more heat than is applied. Perhaps the only affect from varying the applied heat is related to how quickly semi-volatile species evaporate from the sample (higher heat causes them to evolve faster than the lower heat)? As for the pyrolysis phase, how might the heating environment differ between real cooking/heating fires and wildfires compared to these laboratory studies?

4) It was a bit confusing that the graphs with the data as a function of time are not indicative of how much fuel was lost during each phase, which was important for the emission factor calculations. Although using the fuel lost for each phase in the calculation was mentioned in the methods section (and shown in the mass loss plots), it should be repeated again in the text of the emissions section. The other piece of information to include in that section was whether or not the aerosol concentrations were dilution corrected.

5) A short discussion on some (potentially large) sampling issues should be included, especially the effect of dilution on the OA measurements and losses of the fluffy, fractal BC-dominated particles.

6) It is interesting that the organic spectra from the pyrolysis phase contain more alkene and aromatic peaks whereas the smoldering spectra contain more oxygenated peaks. It is quite remarkable that these two types of spectra were also observed in the ambient London data. The dominance of m/z 60 in the smoldering spectra is somewhat surprising since it should be from levoglucosan which is a pyrolysis product from cellulose that forms without oxidation. The rest of the spectra indicate that the OA from this phase is oxidized compared to the OA from the pyrolysis phase. What does this indicate about how m/z 60 is formed?

Minor Comments:

Page 5, line 133: change Appendix 1 to Supplementary Material.

Page 6, Figure 2: The text says that the wood was sanded down and the photo looks like it does not have any bark. Is that the case? Does this fuel have any residual latex that can influence the emissions?

Page 7, lines 197-198: Add further description of the "small flame" that piloted ignition.

Page 8, lines 203-207: All four pieces of wood extinguished at different times. Did they all ignite at the same time?

Page 12, lines 292-295: Could you add the average MCE's from these studies to support these statements?

Page 14, line 337: Should probably be "alkenes, alkanes, cycloalkenes" instead of "alkanes, alkenes, cycloalkanes." Could also mention here that some "saturated hydrocarbon" peaks (e.g., m/z 43) could contain an oxygen.

Page 18, lines 433-435: Where are the qualitative potassium measurements that are referenced here? Maybe include in the SM?

Page 19, line 444: Could mention that Zhou et al. study was a summertime wildfire and the Brito et al. study was of open biomass burning (in contrast to the Young et al study of wintertime heating fires).

Section 3.2 on Nitrogen: Since the organic mass is high, there could be some organic interferences at m/z 30 and 46 that are currently not subtracted in this analysis for organic nitrate. Considering the uncertainties in this calculation, how much of the total mass could have been inorganic nitrate? The next to the last sentence seems to indicate that the ratio of (NO+ plus NO2+) (presumably organic nitrate) to organic mass is 0.29:1. That is quite high. What about other inorganic constituents? There needs to be a conclusion to tie this back to the beginning statements on hygroscopicity.

Table 2: What do the errors indicate? Might want to note in the caption that the emission factors are for the measured mass loss during that phase and a "weighted" average is shown in Figure 7 (although this starts to become clear with Figure 8).

Page 21, line 512: Consider different wording than "both factors."

Page 23, line 555-556: Consider revising "if a flame reached a high intensity and rate of mass loss sooner..." since this phrase is confusing.

Page 24, lines 566-567: The pyrolysis phase as defined in this paper cannot be parameterized with MCE because $CO_2$ and CO are not produced prior to ignition. Consider rephrasing this sentence.

Page 24, line 571. Consider replacing "this third phase" with "the pyrolysis phase."

For all figures: Examine the positons of the text boxes, arrows, and shaded regions to make sure nothing is obscured and all are placed correctly. Consider using three different colors consistently for the three phases, including the bars/shaded regions for Phase 1 and 3 and data points in Figures 6 and 7 (suggest the three colors used for Figure 8).

Figures 3 and S1.1-1.8: Make the CO and MCE traces darker. Make all of the OA traces solid lines (some are dashed).

Figure 4: Change OA and rBC to lines which shows the clear presence/absence of each species instead of filling to zero. Consider omitting the scattering volume from these plots, since they do not appear clearly in the figure nor provide additional information. Why are there multiple Phase 2 regions?

Figure 8 (and related figures in SM): Change x-axis to "mass loss rate", also the associated text in the manuscript should say "rate."
* * *

---

## Author Comment (AC1) · 15 Nov 2017

This work studied aerosol emissions from burning of wood samples from west Africa in a highly-controlled manner. Parameters that were regulated include the shape and mass of wood samples, airflow and surrounding thermal environment. Measurements of OC and rBC emissions were performed online using a compact-AMS and an SP-2 respectively after ~ 100-time dilution. CO and $CO_2$ concentrations were also measured, thus modified combustion efficiency – a qualitative index for combustion condition – was determined. The highly controlled burns coupled with fast real-time measurements made it possible to characterize the fast changes in aerosol emissions over the course of combustion and relate the changes directly to combustion conditions. The connection of the results of this study and those from an ambient study by Young et al. is rather interesting and provides an example for using the results of this study to interpret ambient BBOA factors and differentiate BBOAs emitted from different burning conditions. This is a quality work that generated new results on emissions and characteristics of aerosol from burning and pyrolysis of wood. The manuscript is overall well written and the scope of the work fits well within that of ACP. I recommend acceptance after the authors respond to the following comments.

Speaking of the repeatability of combustion events, which is the core of this work, one question is whether there is quantitative information to demonstrate this. For example, were the same experimental conditions repeated and how the emission rates and aerosol characteristics varied for the same condition? How do the emission factors determined in this study compare with the values reported in literature?

*Perhaps the referee has missed Table 1, which details the conditions of all of the 8 tests carried out. In all cases the infra-red heating was set to be either 30 kW m$^{-2}$ or 50 kW m$^{-2}$ and the air flow rate was set to be either 200 lpm or 50 lpm. These conditions are identified as being low heat (h) or high heat (H) and low flow (f) or high flow (F). The notation used to identify the tests is given in the last row of table 1. Figure 4 and the supplementary figures 1.1-1.8 show how the aerosol characteristics compare for each of these burns. Emission factors for each phase in each test are presented in Table 2, alongside a discussion of these results and comparisons with literature values.*

It was mentioned at the end of page 3 that the usage of a pilot flame had a negligible influence on CO and $CO_2$ emissions. Are there measurements data to demonstrate?

*More detail about the pilot flame has been added in lines 106-109 to back up this statement. The small level of emissions from the pilot flame cannot be detected by the CO/$CO_2$ instruments.*

> A pilot flame was used to initiate flaming combustion, which had a negligible influence on CO and $CO_2$ emissions due to the low flow rate of the pre-mixed flame (< 1.0 lpm) compared with the high flow in the exhaust duct (~150 lps) and the forced airflow imposed in the test chamber (50 or 200 lpm of air). Emissions from the pilot flame would be primarily $CO_2$, but its impact was small enough not to be visible in gas measurements.

Line 164, how was fuel moisture content determined?

*An explanation of this process has been added in lines 179-184.*

> The fuel moisture content (FMC) was approximately 7%, which was calculated on a dry weight basis using the calculation shown in Eq. 1, where $m_{wet}$ and $m_{dry}$ refer to the mass of the pre- and post-treatment wood, respectively. A subsample was conditioned in a convection oven for 24 hours at 60°C, and the weight was measured before and after conditioning.
>
> $$FMC = \frac{m_{wet} - m_{dry}}{m_{dry}}$$

In the calculation of EF in this study, the loss mass was determined by weighing. But how was the effect of the dryness of the fuel accounted for? It would be interesting to also report EFs based on total carbon burned/emitted, e.g., normalized by total carbon measured in CO and CO2. This may allow more direct comparison with field observations.

*The addition of the pyrolysis phase makes this comparison difficult – CO and $CO_2$ aren't released during pyrolysis, so the EF based on CO/$CO_2$ release could only be calculated for phases 2 and 3. Doing this would overlook the large amounts of aerosol released during the pyrolysis phase. This is why these calculations weren't performed in this case.*

Since dilution may change the concentration profile, in Fig. 3, it might be interesting to add the 2nd CO2 measurement data on panel b.

*Unfortunately, our downstream $CO_2$ instrument wasn't working well (see line 124). It was possible to compare the difference in magnitude of abrupt changes, which is how the dilution factor was checked. However, the $CO_2$ time series from the downstream Li-Cor was completely dominated by instrumental drift, which means a direct comparison of the two measurements wasn't possible.*

What's the detection limit of SP2 for rBC? How much of the org/(org+rBC) variation under low rBC conditions shown in Fig. 3c was due to noise in the rBC measurement?

*The detection limit is around 70 nm, (see line 161). A further statement about the effect of noise on the ratio has been added in lines 299-300. As the SP2 is a single particle instrument, there is no latent noise in the measurements (i.e. mass concentrations will never go below zero) only uncertainties generated from counting statistics.*

> In Fig. 3c, grey bars indicate the change in $r_{OA}$ throughout the burn: when this is high, OA dominates emissions; when low, rBC dominates. A ratio is not calculated when the total particulate emissions are less than 0.1 µg m⁻³ due to noise in the signal. Some noise is visible after around 900 s, but this does not fall within the phases studied here so does not affect results.

Line 291, it would be interesting to provide the range of rOA for flaming combustion or quote the "high" and "low" values (and citations) for the subsequent sentence.

*The mean and standard deviation of the $r_{OA}$ values from the studies considered by Reid et al. (2005) has been added in lines 315-6, for comparison.*

Figure 4 is a somewhat difficult to read, I suggest adding an MCE axis in the 2nd and 3 row. Adding ticks on the MCE axis may also be helpful.

*MCE ticks have been added to all graphs.The SP2 scattering readings have been removed, as these were not discussed at length in the text – they can still be found in the supplementary materials. The filled traces have been replaced with lines.*

> Most of the samples were taken from real-world forest or Savanna fires. Reid et al. (2005) reported the ratio of $BC:OA$ from 17 flaming studies (MCE>0.9), from which the $r_{OA}$ can be calculated. In these studies, the mean $r_{OA}$ was 0.87, with a standard deviation of 0.09. The significant difference between this high value and the low one (<0.15) reported in the present study is almost certainly due to the heterogeneous nature of real-world combustion compared with the extremely controlled, high temperature and therefore more efficient combustion that takes place during phase 2 here.

With regard to the paragraph underneath Figure 6: for BBOA, f44 and f43 are not necessarily primarily associated with CO2+ and C2H3O+ respectively. There can be considerable C2H4O+ and C3H7+ in the MS of BBOA.

*Have changed the way $C_3H_7^+$ is referenced in lines 415. High resolution AMS measurements by e.g. Corbin et al. (2015 doi: 10.5194/acp-15-11885-2015) and Ortega et al. (2013 doi: 10.5194/acp-13-11551-2013) show the contribution of $C_2H_4O$ to the m/z 44 peak to be very small in biomass burning.*

> Comparing the relative contributions of significant *m/z* peaks shows further resemblances between the SFOA factors and our results. One example of this is the fractions of *m/z* 44 and 43 (*f*44 and *f*43), which are plotted against one another in Fig. 6*a*. The primary contribution to *f*44 is from the $CO_2^+$ fragment, with a possible contribution from $C_2H_4O^+$; *f*43 is associated with the $C_2H_3O^+$ and $C_3H_7^+$ fragments, which decrease as the aerosol ages.

Bottom paragraph on page 19, although CH2O+ is typically low compared to other organic peaks, since Org/inorg ratio is high in BB aerosol, it can nevertheless be an important contributor to the signal at m/z 30. Organic contribution to m/z 46 can be even more important. These issues should be more clearly discussed since the c-AMS used in this work can't differentiate ions with the same nominal m/z. Also, what was the 46:30 ratio for ammonium nitrate measured during this study?

*We thank the referee for their comments since, on looking into the data again, we have identified an error in the calculation that led to a large upward bias in the nitrate to organic ratio.  This has now been rectified. The results actually show that the most likely contribution to the m/z 30 peak is in fact pure organic. Section 3.1 has been removed and replaced with a paragraph describing this new take on the result in lines 486-96.*

Previous experiments have used the ratio of $m/z$ 46 to $m/z$ 30 to determine the proportion of organic and inorganic nitrate present in aerosol mass (Farmer et al., 2010; Reyes-Villegas et al., 2007). Using data from experiments $hF$.1, $hF$.2 and $HF$.1, a median $m/z$ 46:30 ratio of 0.084 ± 0.15 was found here. Such low ratios have often been used in the literature to suggest the presence of organic nitrate (Hao et al., 2014; Sato et al., 2010; Reyes-Villegas et al., 2017). However, these analyses generally assume only a small organic interference on the $m/z$ 30 peak. Here, the $m/z$ 30 value was low compared with surrounding peaks, which was not the case in the aforementioned studies. An ($m/z$ 30 + $m/z$ 46):OA ratio of 0.029:1 was found. These peaks were emitted concurrently with other organic mass with a correlation of $r^2 = 0.99$. These results strongly suggest that nitrogen-containing species were not emitted in appreciable amounts during this set of experiments.

Line 505 – 508, in addition to the two studies mentioned here, a recent study by Collier et al. (ES&T, 50, 8613–8622, 10.1021/acs.est.6b01617, 2016) also reported a negative correlation between OA emission factors and MCE for wildfires.

*This study has now been added in lines 519-21 and again in lines 547-9.*

[revised manuscript text omitted]

**S4 Alkali metal measurements**

During the flaming period of each test (phase 2), extremely large quantities of potassium were recorded by the AMS at $m/z$ 39 and 41; at times, enough to saturate the instrument. Such measurements are routinely discarded, as the thermal ionisation of alkali metals on the AMS heater result in large, unquantitative measurements. However, qualitatively, measurements of potassium were high during phase 2. Large peaks at $m/z$ 85 and 87 were consistent with the naturally-occurring isotope ratio of rubidium-85 to rubidium-87, suggesting that alkali metals in general were emitted during this phase. The natural and measured abundances of these species are shown in table S1.

**Table S1: Natural and measured abundances of alkali metals.**

| Isotopes | Natural abundance | Measured abundance |
|---|---|---|

| | | |
|---|---|---|
| **Potassium
K-39 : K-41** | 0.932 : 0.067 | 0.934 : 0.066 |
| **Rubidium
Rb-85 : Rb-87** | 0.722 : 0.278 | 0.713 : 0.287 |

---

## Author Comment (AC2) · 15 Nov 2017

Overview: I find this manuscript to be well written and logically organized. The manuscript describes 8 biomass burning measurements of organic aerosol (OA) and refractory black carbon (rBC) from one type of rubberwood material in a highly controlled combustion system. The combustion system was designed through international standards for measuring combustion properties of different fuels for industrial and insurance purposes. This is (at least) one of the first studies that start to bridge the gap between industrial standard combustion practices and ambient environmental studies. I applaud the authors for taking this approach. As these highly detailed measurements are important and there are few such measurements reported in current literature, this manuscript is both timely and appropriate material for ACP. The manuscript should be published with attention paid to the following minor issues.

General comments:

1.) "...from African biomass combustion" in the title implies a broader appeal than the actual use of 32 pieces from one specific type of rubberwood tree. Perhaps, "... from combustion of a common African biofuel source".

*This has been changed.*

2.) What was the mass of the combusted fuel? The only details given are one dimension (100 mm), a picture, and~7% moisture value.

*More detail of the sample mass has been added in lines 178-9.*

> The mass and dimensions of all samples were as uniform as possible: the mean mass of the samples (each containing four pieces of wood) was 160.0 g, with a standard deviation of 12.4 g.

3.) Interestingly, the authors do not emphasize the use of an industrial standard equipment for controlling the combustion process. This is a ready mechanism for others in the field to conduct similar experiments, such that a bit more detail might help move the whole field further along.

*More detail about the reason for conducting experiments as we did, as well as a recognition of the FPA being used a standard methodology for test combustion of materials, has been added in lines 93-100.*

Combustion experiments were carried out in the Rushbrook Fire Laboratory, School of Engineering, University of Edinburgh. The Fire Propagation Apparatus (FPA) allows determination and quantification of material flammability characteristics including time to ignition, gaseous emissions, burning rate and heat release rate. Small (approximately 100 mm) samples of fuel are heated and ignited under highly-controlled conditions (Brohez et al., 2006). Emissions from combustion are inherently linked to the combustion environment, which adds a layer of complexity for free burning fires as the number of variables impacting the burning behaviour and emissions increases. Therefore, in order to understand emissions more clearly, it is important to understand the impact of the test environment on the combustion processes. Essentially, the FPA is a reactor chamber in which it is possible to study the burning behaviour of materials while maximising control over important parameters, but without diverting too far from a free-burning fire. Use of the FPA is widespread as a standard methodology for test combustion of materials for approval in industrial applications, as it provides highly repeatable conditions. Such conditions have not, to date, been available often in environmental burn studies.

4.) In the discussion of the OA mass spectra, is there a reason why PMF was not done on the current data set to see how well it reproduces the average spectra for the two conditions (phase 1 and 3)?

*PMF is a method for deconstructing complex datasets that result from the superposition of different sources. In this case, the source was already isolated. As such, it was not felt that the use of PMF in this case would contribute towards the scientific discussion.*

5.) In the discussion of Figure 6, placing either PMF factors or phase 1/3 weighted means on the plots would help in interpreting the discussions of the relative abundances of f44 vs f43 and f44 vs f60. For example, line 395 "On average, the f44 contribution increased as combustion developed" is stated without an average value on the plot to assess. Attempting to discern "averages" of data points on a plot is difficult for readers, especially when the markers overlap.

*Median values for the emissions during phases 1 and 3 have been added to both plots in Fig. 6.*

[Figure]

Figure 6: f44 vs f43 (a) and f44 vs f60 (b), with phase 1 data shown in purple and phase 3 in green. **The larger, outlined points show the median values for each phase.** The general trend of results from Heringa et al. (2012) is illustrated by the black arrow in (a) and results from SFOA2 and SFOA1 collected by Young et al. (2015) are reproduced in both plots.

6.) Differences in the PMF factors from Young et al. showed very nice correlations with the differences in the mass spectra from phase 1 and 3. How do the actual PMF factors compare with the phase 1/3 mass spectra? While the differences are important, the actual connection in the text is that the Young et al. SFOA1 and SFOA2 may well represent phases 1/3 in ambient, which is a comparison of the absolute individual spectra, rather than just differences.

*We did examine the absolute individual mass spectra, but we have focused the discussion on the difference between the two mass spectra. The two ambient spectra reported by Young et al. are not as distinct as the two spectra reported here. This is both because the real-world combustion sources will be a complex mixture of both emission types, and because ambient PMF data is separated by temporal trends as well as spectral trends, so a less distinct separation would be expected.*

7.) During the discussion of ambient measurements (around lines 505 and 560+), the authors cite Zhou et al (2017) and Brito et al (2014), but appear to have missed Collier et al (2016), which showed a very strong correlation between OA/(CO2+CO) (which is a type of ER or emission ratio, rather than EF or emission factor, given the lack of information on fuel mass) and MCE in ambient wildland fires in the western US. The current lab results and these ambient results are in strong agreement about how OA emissions are governed in large part by combustion conditions for similar fuel types. They may differ by the pyrolysis emissions and/or effects of heat. The implications emphasized herein and noted in Collier et al. are important from both ambient and lab viewpoints and could readily be included.

*The Collier et al. (2016) paper has now been added in lines 519-21, and referenced again in lines 547-9.*

A similar result was noted by Weimer et al. (2008), who observed a distinct difference between particulate emissions from different phases, but almost none between wood types. Collier et al. (2016) observed considerably higher OA emissions during smouldering than flaming emissions during measurements of wildfires in the Pacific Northwest region of the United States. This is supported here: the greatest differences in particle emissions were between individual phases of combustion, rather than being between different burning environments.

8.) Lines 483-485. The authors appear to give a median values of m/z 46:30 as 0.084+/= 0.15 and linear correlation with a slope of 0.29. Are these values from the same distributions, suggesting that the m/z 46:30 distribution is highly skewed? What does a skewed distribution say about the results?

*We thank the referee for their comments since, on looking into the data again, we have identified an error in the calculation that led to a large upward bias in the nitrate to organic ratio. This has now been rectified. The results actually show that the most likely contribution to the* m/z *30 peak is in fact pure organic. Section 3.1 has been removed and replaced with a paragraph describing this new take on the result in lines 486-96.*

Previous experiments have used the ratio of *m/z* 46 to *m/z* 30 to determine the proportion of organic and inorganic nitrate present in aerosol mass (Farmer et al., 2010; Reyes-Villegas et al., 2007). Using data from experiments *hF*.1, *hF*.2 and *HF*.1, a median *m/z* 46:30 ratio of 0.084 ± 0.15 was found here. Such low ratios have often been used in the literature to suggest the presence of organic nitrate (Hao et al., 2014; Sato et al., 2010; Reyes-Villegas et al., 2017). However, these analyses generally assume only a small organic interference on the *m/z* 30 peak. Here, the *m/z* 30 value was low compared with surrounding peaks, which was not the case in the aforementioned studies. An (*m/z* 30 + *m/z* 46):OA ratio of 0.029:1 was found. These peaks were emitted concurrently with other organic mass with a correlation of $r^2$ = 0.99. These results strongly suggest that nitrogen-containing species were not emitted in appreciable amounts during this set of experiments.

9.) The authors rightly emphasize the potential impact of pyrolysis OA emissions in the underlying emission factor variations observed with MCE (especially lab studies); however, the authors thereby downplay differences in heat load on fuels for a given fire. Here, the authors include this parameter explicitly (perhaps for the first time for ambient-related measurements). The heat load can vary depending upon whether the combustion occurs inside a stove (i.e., solid fuel combustion) or open air (wild land combustion). It can also vary by how much fuel is available (both in lab and ambient) and how well packed. Might this have a large an impact on measured OA emissions per MCE, similar to that suggested for pyrolysis?

*A comment on this has been added in lines 588-92.*

This could begin to explain some of the large variability in biomass burning EF measurements, even between tests with similar MCEs and within the same investigation. If this is the case, identifying a further parameter that is able to capture pyrolysis OA could add a significant further constraint to bottom-up studies of EFs. Furthermore, the heat load placed on samples, which was included in this study as a specific parameter, was found to influence the quantity of OA emitted during pyrolysis. In real-world combustion, the heat load can vary substantially based on how the fuel is packed and where combustion is taking place. Hence, it can be very different between wood stove burns and wildfires, and vary according to burn conditions or fuel density in each type of combustion.

Reviewer 1: purple                    Reviewer 2: blue                    Reviewer 3: red

[revised manuscript text omitted]

Reviewer 1: purple          Reviewer 2: blue          Reviewer 3: red

Farmer, D. K., Matsunaga, A., Docherty, K. S., Surratt, J. D., Seinfeld, J. H., Ziemann, P. J. and Jimenez, J. L.: Response of an aerosol mass spectrometer to organonitrates and organosulfates and implications for atmospheric chemistry, P. Natl. Acad. Sci. U.S.A., 107, 6670-6675, doi: 10.1073/pnas.0912340107, 2010.

Ferek, R. J., Reid, J. S., Hobbs, P. V., Blake, D. R., and Liousse, C.: Emission factors of hydrocarbons, halocarbons, trace gases, and particles from biomass burning in Brazil, J. Geophys. Res., 103(D24), 32107–32118, doi:10.1029/98JD00692, 1998.

Freeborn, P. H., Wooster, M. J., Hao, W. M., Ryan, C. A., Nordgren, B. L., Baker, S. P. and Ichoku, C.: Relationships between energy release, fuel mass loss, and trace gas and aerosol emissions during laboratory biomass fires, J. Geophys. Res., 113(D1), doi: 10.1029/2007JD008679, 2008.

~~Fry, J. L., Kiendler-Scharr, A., Rollins, A. W., Wooldridge, P. J., Brown, S. S., Fuchs, H., Dubé, W., Mensah, A., dal Maso, M., Tillmann, R., Dorn, H. P., Brauers, T. and Cohen, R. C.: Organic nitrate and secondary organic aerosol yield from no3 oxidation of β-pinene evaluated using a gas-phase kinetics/aerosol partitioning model, Atmos. Chem. Phys., 9, 1431-1449, doi: 10.5194/acp-9-1431-2009, 2009.~~

Hao, L. Q., Kortelainen, A., Romakkaniemi, S., Portin, H., Jaatinen, A., Leskinen, A., Komppula, M., Miettinen, P., 750   Sueper, D., Pajunoja, A., Smith, J. N., Lehtinen, K. E. J., Worsnop, D. R., Laaksonen, A. and Virtanen, A.: Atmospheric submicron aerosol composition and particulate organic nitrate formation in a boreal forestland-urban mixed region, Atmos. Chem. Phys., 14, 13483-13495, doi: 10.5194/acp-14-13483-2014, 2014.

Harrison, R. M., Beddows, D. C. S., Hu, L. and Yin, J.: Comparison of methods for evaluation of wood smoke and estimation of UK ambient concentrations, Atmos. Chem. Phys., 12, 8271-8283, doi: 10.5194/acp-12-8271-2012, 755   2012.

Heringa, M. F., DeCarlo, P. F., Chirico, R., Lauber, A., Doberer, A., Good, J., Nussbaumer, T., Keller, A., Burtscher, H., Richard, A., Miljevic, B., Prevot, A. S. H. and Baltensperger, U.: Time-resolved characterization of primary emissions from residential wood combustion appliances, Environ. Sci. Technol., 46, 11418-11425, doi: 10.1021/es301654w, 2012.

Hobbs, P., Reid, J., Kotchenruther, R. A., Ferek, R. J. and Weiss, R.: Direct radiative forcing by smoke from biomass burning, Science, 275(5307), 1777-1778, doi: 10.1126/science.275.5307.1777 , 1997.

Janhäll, S., Andreae, M. O. and Pöschl, U.: Biomass burning aerosol emissions from vegetation fires: particle number and mass emission factors and size distributions, Atmos. Chem. Phys., 10, 1427-1439, doi: 10.5194/acp-10-1427-2010, 2010.

Johansson, L. S., Leckner, B., Gustavsson, L., Cooper, D., Tullin, C. and Potter, A.: Emission characteristics of modern and old-type residential boilers fired with wood logs and wood pellets, Atmos. Environ., 38(25), 4183-4195, 770   doi: 10.1016/j.atmosenv.2004.04.020, 2004.

Reviewer 1: purple                 Reviewer 2: blue                 Reviewer 3: red

[revised manuscript text omitted]

**S3 OA and rBC concentrations vs mass loss for each experiment carried out, averaged over 5 s.**

[Figure]

Figure S2: OA emissions vs mass loss for each experiment, averaged over every 5 seconds.

[Figure]

**Figure S3: rBC emissions vs mass loss for each experiment, averaged over every 5 seconds.**

**S4 Alkali metal measurements**

During the flaming period of each test (phase 2), extremely large quantities of potassium were recorded by the AMS at $m/z$ 39 and 41; at times, enough to saturate the instrument. Such measurements are routinely discarded, as the thermal ionisation of alkali metals on the AMS heater result in large, unquantitative measurements. However, qualitatively, measurements of potassium were high during phase 2. Large peaks at $m/z$ 85 and 87 were consistent with the naturally-occurring isotope ratio of rubidium-85 to rubidium-87, suggesting that alkali metals in general were emitted during this phase. The natural and measured abundances of these species are shown in table S1.

**Table S1: Natural and measured abundances of alkali metals.**

| Isotopes | Natural abundance | Measured abundance |
|---|---|---|

| | | |
|---|---|---|
| **Potassium**
**K-39 : K-41** | 0.932 : 0.067 | 0.934 : 0.066 |
| **Rubidium**
**Rb-85 : Rb-87** | 0.722 : 0.278 | 0.713 : 0.287 |

---

## Author Comment (AC3) · 15 Nov 2017

This paper reports organic aerosol (OA) and refractory black carbon (rBC) emissions along with gas phase carbon dioxide ($CO_2$) and carbon monoxide (CO) emissions from controlled laboratory experiments where the same amount and type of fuel was burned with varying applied heat and flow rates of air through the sample. The results are relevant to the rapidly growing body of work on aerosol emissions from biomass burning. A major finding of this work is that there are distinct types of aerosols from each of the three different phases of the controlled fires: predominantly less-oxidized OA during the initial pyrolysis phase (Phase 1, prior to ignition with no emissions of $CO_2$ or CO), almost entirely rBC during the flaming phase (Phase 2, after ignition with a relatively high modified combustion efficiency or MCE), and a mixture of mostly more oxidized OA with a small amount of rBC during the smoldering phase (Phase 3 after ignition with a relatively low MCE). The relative amounts of aerosols between the three phases were generally consistent between experiments with the same heat+flowrate conditions. The results of these laboratory experiments were compared with other laboratory and field measurements.

The manuscript is fairly clear to understand, but some of the analysis and conclusions are confusing given the types of experiments conducted. This paper is important to many readers of Atmospheric Chemistry and Physics and should be published after some clarifications are made to the manuscript. The key areas needing revisions are described below.

Overall Comments:

1) The manuscript is lacking a conceptual picture of the processes examined in these experiments, and interpreting the results depends on such a foundation. For this study, the heat applied to the sample (high heat = H or low heat = h) and the flow rate across the sample (high flow = F or low flow = f) were varied for 8 experiments. How should the results vary with these changing experimental conditions? It is likely dependent on the applied heat and flow rate in different ways for the different phases.

2) Prior to ignition, it is expected that the amount of heat applied and sample flow rate will affect the amount of vapors up to their flashpoint (ignition) as well as the amount available for subsequent re-condensation as OA. Higher heat applied increases the vaporization rate of semi-volatile species from the sample. Higher/lower flow rate would then decrease/increase the concentration just prior to ignition. rBC is not expected to be formed during the pyrolysis phase since there is no combustion occurring (no $CO_2$ or CO). What would happen if an ignition source was not there? Since this is an important phase for producing OA and is not a combustion phase, how might the fuel type be relevant for the emissions of this phase? What kind of differences might be expected between wood used as cooking or heating fuel versus wildfires? A little more discussion on this subject should be included in the paper.

3) After ignition, how does the applied heat compare with the heat produced by combustion for the flaming and smoldering phases? Applied heat may be irrelevant to the results if the combustion processes generate more heat than is applied. Perhaps the only affect from varying the applied heat is related to how quickly semi-volatile species evaporate from the sample (higher heat causes them to evolve faster than the lower heat)? As for the pyrolysis phase, how might the heating environment differ between real cooking/heating fires and wildfires compared to these laboratory studies?

*These three comments focus on the conceptual understanding of the processes involved in these experiments. It is beyond the scope of this work to definitively provide an underlying physical basis for real world emissions, although future research using a similar method and a larger range of conditions could perhaps do this.*

*Some extra comments on the differences between results under different conditions has been included in lines 344-5 and 352-5.*

*We agree that addressing the links between these experiments and real-world combustion more directly is important. A short discussion has been included in the summary section (lines 628-37), which seeks to relate our results to the likely differences in wildfire and cookstove burning environments.*

The presentation of different chemical characteristics for three phases of biomass burning here provides a physical basis that could help to explain why there is such large variability in biomass burning emissions. This variability creates difficulties when producing bottom-up representations of emissions. If the difference in emissions associated with the pyrolysis and smouldering phases are better understood, however, and if further approaches to quantifying pyrolysis are developed, it could be possible to constrain emission estimates more completely. The development of real-world combustion differs from our experimental burns, which may affect the balance of emissions in the different phases. The yield of OA from the pyrolysis phase was largely dictated by the heat that reached the sample, with higher applied heat increasing the vaporisation of semi-volatile species. In the context of a wildfire, a number of variables could affect the rapidity and intensity of heating, and hence the pyrolysis yield. Examples of these include the fuel density, the fuel water content or the radiant heating of unburnt fuel adjacent to flaming combustion. The high radiant heat in the centre of a wildfire could show features of our high heat experiments, although this could be hindered by the fuel type or moisture content. Conditions are more constrained in a stove used for cooking or heating, where fuel is introduced in batches, but variations in fuel loading, moisture and type still result in more variation in emissions than has been seen here.

4) It was a bit confusing that the graphs with the data as a function of time are not indicative of how much fuel was lost during each phase, which was important for the emission factor calculations. Although using the fuel lost for each phase in the calculation was mentioned in the methods section (and shown in the mass loss plots), it should be repeated again in the text of the emissions section. The other piece of information to include in that section was whether or not the aerosol concentrations were dilution corrected.

*The rate of mass loss has been added to Fig. 3 and the related supplementary plots. An in-text reference to the method of EF calculation and the dilution correction has been added in lines 502-4.*

> In order to compare results collected here with those in the literature, average EFs for the duration of each phase of combustion have been calculated for OA and rBC for each of the experiments. These have been calculated based on aerosol emissions and the measured mass loss during each phase, as outlined in section 2, with a correction of 100 made to account for dilution in the aerosol sample line.

5) A short discussion on some (potentially large) sampling issues should be included, especially the effect of dilutionon the OA measurements and losses ofthe fluffy, fractal BC-dominated particles.

*Details of potential sampling issues has been included in the manuscript in lines 129-132.*

> The diluters used have been designed to sample combustion particles and therefore to minimise particle losses. However, it must be recognised that there may be some perturbation of aerosol composition through processes such as the repartitioning of OA species or diffusional loss of smaller particles. The experiment design has minimised these effects as much as possible.

6)It is interesting that the organic spectra from the pyrolysis phase contain more alkene and aromatic peaks whereas the smoldering spectra contain more oxygenated peaks. It is quite remarkable that these two types of spectra were also observed in the ambient London data. The dominance of m/z60 in the smoldering spectra is some what surprising since it should be from levoglucosan which is a pyrolysis product from cellulose that forms without oxidation. The rest of the spectra indicate that the OA from this phase is oxidized compared to the OA from the pyrolysis phase. What does this indicate about how m/z 60 is formed?

*Technically, all organic aerosol emissions measured here are forms of pyrolysis. However, phase 1 takes place at a lower temperature and in the absence of combustion. Levoglucosan is a pyrolysis product of cellulose, which pyrolyses at relatively high temperatures and so this manifests more strongly during phase 3.*

Minor Comments:

Page 5, line 133: change Appendix 1 to Supplementary Material.

*This has been changed.*

Page 6, Figure 2: The text says that the wood was sanded down and the photo looks like it does not have any bark. Is that the case? Does this fuel have any residual latex that can influence the emissions?

*There was no bark on the wood – this has now been added to lines 177-8.*

> Each piece was cut from one of three similar lengths of wood with a diameter of approximately 10 cm, before being sanded down to ensure the highest possible consistency across the separate tests (see Fig. 2); bark was removed during the sanding process.

Page 7, lines 197-198: Add further description of the "small flame" that piloted ignition.

*More detail has been added in lines 218-9.*

> The pilot flame was a pre-mixed flame made from an ethylene/air mixture, which formed a blue conical flame of around 1 mm in diameter and 1-1.5 mm in length.

Page 8, lines 203-207: All four pieces of wood extinguished at different times. Did they all ignite at the same time?

*Yes – there was a general build-up of pyrolysis gases before ignition, which wasn't localised to any single piece of wood.*

Page 12, lines 292-295: Could you add the average MCE's from these studies to support these statements?

*Added to line 315.*

> Most of the samples were taken from real-world forest or Savanna fires. Reid et al. (2005) reported the ratio of BC:OA from 17 flaming studies (MCE>0.9), from which the $r_{OA}$ can be calculated. In these studies, the mean $r_{OA}$ was 0.87, with a standard deviation of 0.09.

Page 14, line 337: Should probably be "alkenes, alkanes, cycloalkenes" instead of "alkanes, alkenes, cycloalkanes." Could also mention here that some "saturated hydrocarbon" peaks (e.g., m/z 43) could contain an oxygen.

*Alkanes and alkenes have been switched. A comment on oxygen has been added from lines 361-2.*

> These peaks are associated with fragments of saturated alkenes, alkanes, cycloalkanes and aromatic compounds, respectively, although it is possible that some of these peaks (e.g. *m/z* 43) could contain an oxygen.

Page 18, lines 433-435: Where are the qualitative potassium measurements that are referenced here? Maybe include in the SM?

*A section has now been added to the SM and it has been referenced in line 462-3.*

Page 19, line 444: Could mention that Zhou et al. study was a summertime wildfire and the Brito et al. study was of open biomass burning (in contrast to the Young et al study of wintertime heating fires).

*More detail has been added to lines 479-82.*

This suggests that results presented here are likely to be most valuable in situations where measurements are made in close proximity to the biomass burning source and where atmospheric processing of aerosol is limited. Zhou et al. (2017)'s study was of a summertime wildfire, and Brito et al. (2014)'s of open biomass burning, in contrast to Young et al. (2015)'s observations, which were identified with household heating fires. Thus, it is possible that the nature of the combustion in these different circumstances played a role in the emissions released.

Section 3.2 on Nitrogen: Since the organic mass is high, there could be some organic interferences at m/z 30 and 46 that are currently not subtracted in this analysis for organic nitrate. Considering the uncertainties in this calculation, how much of the total mass could have been inorganic nitrate? The next to the last sentence seems to indicate that the ratio of (NO+ plus NO2+) (presumably organic nitrate) to organic mass is 0.29:1. That is quite high. What about other inorganic constituents? There needs to be a conclusion to tie this back to the beginning statements on hygroscopicity.

*We thank the referee for their comments since on looking into the data again we have identified an error in the calculation which led to a large upward bias in the nitrate to organic ration. This has now been rectified. The results actually show that the most likely contribution to the m/z 30 peak is in fact pure organic. Section 3.1 has been removed and replaced with a paragraph describing this new take on the result in lines 486-496.*

Previous experiments have used the ratio of $m/z$ 46 to $m/z$ 30 to determine the proportion of organic and inorganic nitrate present in aerosol mass (Farmer et al., 2010; Reyes-Villegas et al., 2007). Using data from experiments $hF.1$, $hF.2$ and $HF.1$, a median $m/z$ 46:30 ratio of 0.084 ± 0.15 was found here. Such low ratios have often been used in the literature to suggest the presence of organic nitrate (Hao et al., 2014; Sato et al., 2010; Reyes-Villegas et al., 2017). However, these analyses generally assume only a small organic interference on the $m/z$ 30 peak. Here, the $m/z$ 30 value was low compared with surrounding peaks, which was not the case in the aforementioned studies. An ($m/z$ 30 + $m/z$ 46):OA ratio of 0.029:1 was found. These peaks were emitted concurrently with other organic mass with a correlation of $r^2$ = 0.99. These results strongly suggest that nitrogen-containing species were not emitted in appreciable amounts during this set of experiments.

Table 2: What do the errors indicate? Might want to note in the caption that the emission factors are for the measured mass loss during that phase and a "weighted" average is shown in Figure 7 (although this starts to become clear with Figure 8).

*Errors indicate the standard error of the means (SD/sqrt(n)), taking account of covariance. This has been added to the caption.*

Page 21, line 512: Consider different wording than "both factors."

*Now line 524.*

> Considering the entire combustion period instead of individual phases, Fig. 7a shows a negative correlation between the overall OA emission factor and MCE, with both being strongly influenced by the burning environment in which combustion took place.

Page 23, line 555-556: Consider revising "if a flame reached a high intensity and rate of mass loss sooner..." since this phrase is confusing.

*Done. Now lines 570-1.*

> In Fig. 8 this transition can be seen at a rate of mass loss of approximately 0.15 g s$^{-1}$, after which the gradient becomes noticeably steeper. Thus, disproportionately more rBC was produced by flames that remained above this threshold mass loss rate for longer.

Page24, lines 566-567: The pyrolysis phase as defined in this paper cannot be parameterized with MCE because CO2 and CO are not produced prior to ignition. Consider rephrasing this sentence.

*Done. Now lines 580-1.*

> The pyrolysis phase as defined here cannot be parameterised with MCE, as CO and CO$_2$ are not produced prior to ignition. Nevertheless, both the large concentrations of OA released during phase 1 here, and the comparison with Young et al. (2015)'s ambient results shown in Sect. 3.1, suggest that direct pyrolysis emissions could account for a significant proportion of ambient BBOA aerosol described in the literature.

Page 24, line 571. Consider replacing "this third phase" with "the pyrolysis phase."

*Done. Now line 585.*

> It is therefore possible that the current approach of describing biomass burning emissions using mass loss and MCE as key variables is unable to account for emissions released during pyrolysis.

For all figures: Examine the positons of the text boxes, arrows, and shaded regions to make sure nothing is obscured and all are placed correctly. Consider using three different colors consistently for the three

phases, including the bars/shaded regions for Phase 1 and 3 and data points in Figures 6 and 7 (suggest the three colors used for Figure 8).

*Thanks to the referee for this good idea – this is now updated in Fig. 3, Fig. 4, Fig. 6, and supplementary figures Fig. S1.1-8. Positioning of text boxes etc has been checked throughout. In Fig. 7, data is presented as overall averages from each test, so phase colours can't be used.*

Figures 3 and S1.1-1.8: Make the CO and MCE traces darker. Make all of the OA traces solid lines (some are dashed).

*The first subplot has now been divided into two for clarity, and a trace for mass loss rate added. Colours are darker and OA traces are all now solid.*

[Figure]

Figure 4: Change OA and rBC to lines which shows the clear presence/absence of each species instead of filling to zero. Consider omitting the scattering volume from these plots, since they do not appear clearly in the figure nor provide additional information.

*All of these have been done. SP2 scattering information is still available in the supplementary figures S1.1-1.8.*

Why are there multiple Phase 2 regions?

*These have been removed as they were confusing.*

Figure 8 (and related figures in SM): Change x-axis to "mass loss rate", also the associated text in the manuscript should say "rate."

*This has been updated.*

Reviewer 1: purple          Reviewer 2: blue          Reviewer 3: red

[revised manuscript text omitted]

Reviewer 1: purple          Reviewer 2: blue          Reviewer 3: red

[revised manuscript text omitted]

**S4 Alkali metal measurements**

During the flaming period of each test (phase 2), extremely large quantities of potassium were recorded by the AMS at $m/z$ 39 and 41; at times, enough to saturate the instrument. Such measurements are routinely discarded, as the thermal ionisation of alkali metals on the AMS heater result in large, unquantitative measurements. However, qualitatively, measurements of potassium were high during phase 2. Large peaks at $m/z$ 85 and 87 were consistent with the naturally-occurring isotope ratio of rubidium-85 to rubidium-87, suggesting that alkali metals in general were emitted during this phase. The natural and measured abundances of these species are shown in table S1.

**Table S1: Natural and measured abundances of alkali metals.**

| Isotopes | Natural abundance | Measured abundance |
| --- | --- | --- |

| | | |
|---|---|---|
| **Potassium K-39 : K-41** | 0.932 : 0.067 | 0.934 : 0.066 |
| **Rubidium Rb-85 : Rb-87** | 0.722 : 0.278 | 0.713 : 0.287 |